# Non-associative phase separation in an evaporating droplet as a model for prebiotic compartmentalization

Wei Guo[1,5], Andrew B. Kinghorn [2,5], Yage Zhang [1], Qingchuan Li[1,3], Aditi Dey Poonam[1], Julian A. Tanner [2,4✉] & Ho Cheung Shum [1,4✉]

The synthetic pathways of life's building blocks are envisaged to be through a series of complex prebiotic reactions and processes. However, the strategy to compartmentalize and concentrate biopolymers under prebiotic conditions remains elusive. Liquid-liquid phase separation is a mechanism by which membraneless organelles form inside cells, and has been hypothesized as a potential mechanism for prebiotic compartmentalization. Associative phase separation of oppositely charged species has been shown to partition RNA, but the strongly negative charge exhibited by RNA suggests that RNA-polycation interactions could inhibit RNA folding and its functioning inside the coacervates. Here, we present a prebiotically plausible pathway for non-associative phase separation within an evaporating all-aqueous sessile droplet. We quantitatively investigate the kinetic pathway of phase separation triggered by the non-uniform evaporation rate, together with the Marangoni flow-driven hydrodynamics inside the sessile droplet. With the ability to undergo liquid-liquid phase separation, the drying droplets provide a robust mechanism for formation of prebiotic membraneless compartments, as demonstrated by localization and storage of nucleic acids, in vitro transcription, as well as a three-fold enhancement of ribozyme activity. The compartmentalization mechanism illustrated in this model system is feasible on wet organophilic silica-rich surfaces during early molecular evolution.

---

[1] Department of Mechanical Engineering, Faculty of Engineering, The University of Hong Kong, Hong Kong (SAR), Hong Kong, China. [2] School of Biomedical Sciences, LKS Faculty of Medicine, The University of Hong Kong, Hong Kong (SAR), Hong Kong, China. [3] School of Chemistry & Chemical Engineering, National Engineering Research Center for Colloidal Materials, Shandong University, Jinan 250100, China. [4] Advanced Biomedical Instrumentation Centre, Hong Kong Science Park, Shatin, New Territories, Hong Kong (SAR), Hong Kong, China. [5] These authors contributed equally: Wei Guo, Andrew B. Kinghorn. ✉email: jatanner@hku.hk; ashum@hku.hk

Microdroplets have been demonstrated as prebiotically plausible reactors[1–3]. Microdroplet reactors facilitate thermodynamically unfavorable synthetic reactions in bulk solution, where one product molecule is formed from two or more reactant molecules with a high loss of entropy[4]. In particular, evaporation could have played a critical role in these prebiotically relevant processes, such as concentrating reagents to enhance the reaction rates inside microdroplets[5–7], and regulating conditions for biopolymer synthesis through wet-dry cycling[8–10]. Despite overcoming the inherent thermodynamic unfavorability in chemical synthesis, how prebiotically synthesized polymers assemble into cellular and subcellular-like compartments capable of autocatalysis[11] and self-replication[12,13] remains unresolved.

Liquid-liquid phase separation (LLPS) has recently been shown as the route for the formation of intracellular membraneless organelles[14,15]. Besides their biological functions in modern cells, the existence of membraneless organelles in cells provides a new perspective on prebiotic compartments on the early Earth[16]. Such non-membrane-bound compartments may represent "remnants" of ancient structures for spatiotemporally regulated biochemical reactions[17]. This concept dates back to Alexander Oparin's arguments in the 1930s[18]. LLPS is a thermodynamically transient, nonequilibrium process leading to phase segregation (nonassociative or segregative phase separation) or the formation of coacervates (associative phase separation) in an initially well-mixed macromolecular-containing system. LLPS can be tuned by varying polymer concentration, charge density, pH, and temperature. With the ability to concentrate biopolymers and form primordial compartments in the dilute "primordial soup"[19], LLPS has been reported to guide the transition from prebiotically synthesized polymers to highly organized structures. This may have driven the evolutionary engine of the first living cells under prebiotic conditions[20].

Currently, most proposed prebiotically plausible membraneless compartments are poly-electrolyte-rich coacervate droplets formed by associative LLPS triggered by specific interactions (such as electrostatics, cation–π interactions, dipole–dipole contacts as well as π–π stacking[21]) of oppositely charged polymers. A unique advantage of coacervate droplet formation as a prebiotic pathway of compartmentalization is its spontaneous assembly in dilute solution. Coacervate droplets could help transition from prebiotically synthesized biopolymers to fatty acid membranes encapsulated protocells[22–24]. However, limitations still exist for complex coacervates as prebiotic reactors. Limitations include the structural stability and inhibited function of RNA due to the complex ion-pairing interactions between RNA and polycations in coacervate droplets. For example, RNA polymerization has shown to be inhibited in polyamine-containing coacervates[25], and the rate constant of RNA cleavage is 60-fold slower inside complex coacervates than that in buffers[26]. In contrast, a similar RNA catalytic reaction exhibits a 70-fold increased rate of reaction through segregative phase separation of polyethylene glycol (PEG)/dextran aqueous two-phase system (ATPS)[27]. It has also been reported that nucleic acid duplexes can be destabilized inside complex coacervate droplets[28]. Hence, a prebiotically plausible pathway for nonassociative phase separation of two neutral polymers would provide an important missing piece of the puzzle for prebiotic compartmentalization. However, the prebiotic pathway for emergence of nonassociative phase separation faces a major challenge to reach such high polymer concentrations in the dilute and homogeneous "primordial soup"[29].

In the free energy landscape, nonassociative phase separation occurs when mixing entropy is overcome by nonspecific molecular interactions and thus polymer incompatibility appears in solution[30,31]. Hence, to trigger nonassociative phase separation in dilute solution, additional nonequilibrium settings (NES) must be introduced to enhance nonspecific molecular interactions or reduce mixing entropy. NES has been widely applied in prebiotic chemistry to maintain thermodynamically unfavored processes such as accumulation of biomolecules[32–34]. A recent example is that, to mimic molecular evolution within porous volcanic rocks on the early Earth, heated microbubbles were used as reactors for assembly and localization of prebiotic molecules[35]. Nevertheless, despite segregative LLPS of ATPS being hypothesized to play important roles in origins of life chemistry[27,29,36], how nonassociative phase separation can be achieved under prebiotically plausible NES remains largely unexplored.

Inspired by the widely introduced evaporation process in wet-dry cycling and prebiotically plausible microdroplet reactors, here we report droplet evaporation-assisted nonassociative phase separation for prebiotic compartmentalization inside a sessile droplet of ATPS. By formulating the LLPS mechanism under the NES, we show that the nonuniform evaporation flux and initial composition of the sessile droplets determine the formation and evolution of self-organized membraneless compartments. The evolution of phase-separated membraneless compartments inside the aqueous sessile droplet looks and behaves similar to intracellular phase separation. Most importantly, the membraneless compartments formed in our model system can serve as spatially functional reactors for enrichment of informational molecules and flow of genetic information, validated by in vitro transcription (IVT) and enhanced ribozyme activities. Our model can validate nonassociative phase separation in moderate conditions with no heating, cooling, or osmotic shock needed. Thus, we elucidate a plausible pathway for prebiotic compartmentalization via nonassociative phase separation. Furthermore, the simple yet robust system can capture both the macromolecularly crowded interior environment and the nonequilibrium features of extant cells, providing cell-mimicking conditions essential for a variety of intracellular functions.

## Results

**Nonassociative phase separation inside the evaporating sessile droplet.** A single-phase mixture of PEG and dextran, which has been extensively studied in our previous work[37–40], is prepared and used as the stock solution for nonassociative phase separation. To achieve the NES required for LLPS, a droplet extracted from the stock solution is pipetted onto a glass slide then it starts to evaporate. Similar scenarios with such NES, that aqueous sessile droplets dispersed on the organophilic silica-rich surfaces, could be found ubiquitously on the early Earth (Fig. 1a), which is of essential significance for prebiotic polymerization of long oligomers[41–43]. Moreover, it has recently been demonstrated that the evaporation on wave-wetted rocks near carbonate-rich lakes would increase phosphate concentration to the molar level, a critical concentration for laboratory prebiotic synthesis[44]. Hence, by sharing similar prebiotically plausible conditions mentioned above, we show that nonassociative phase separation for polymer self-organization and compartmentalization could also be achieved through droplet evaporation (Fig. 1a).

Figure 1c and d show the image sequence of phase separation and polymer self-organization within the evaporating sessile droplet. Initially, the sessile droplet exhibits a homogeneous fluorescence intensity, indicating a single-phase mixture and a lack of compartments inside the droplet. Over time, the green fluorescence domains gradually appear, indicating the occurrence of phase separation and formation of the dextran-rich compartments. With different initial concentrations of polymers dissolved in the droplet, there are two distinct morphologies of self-organized patterns of polymer phase separation, as shown by the

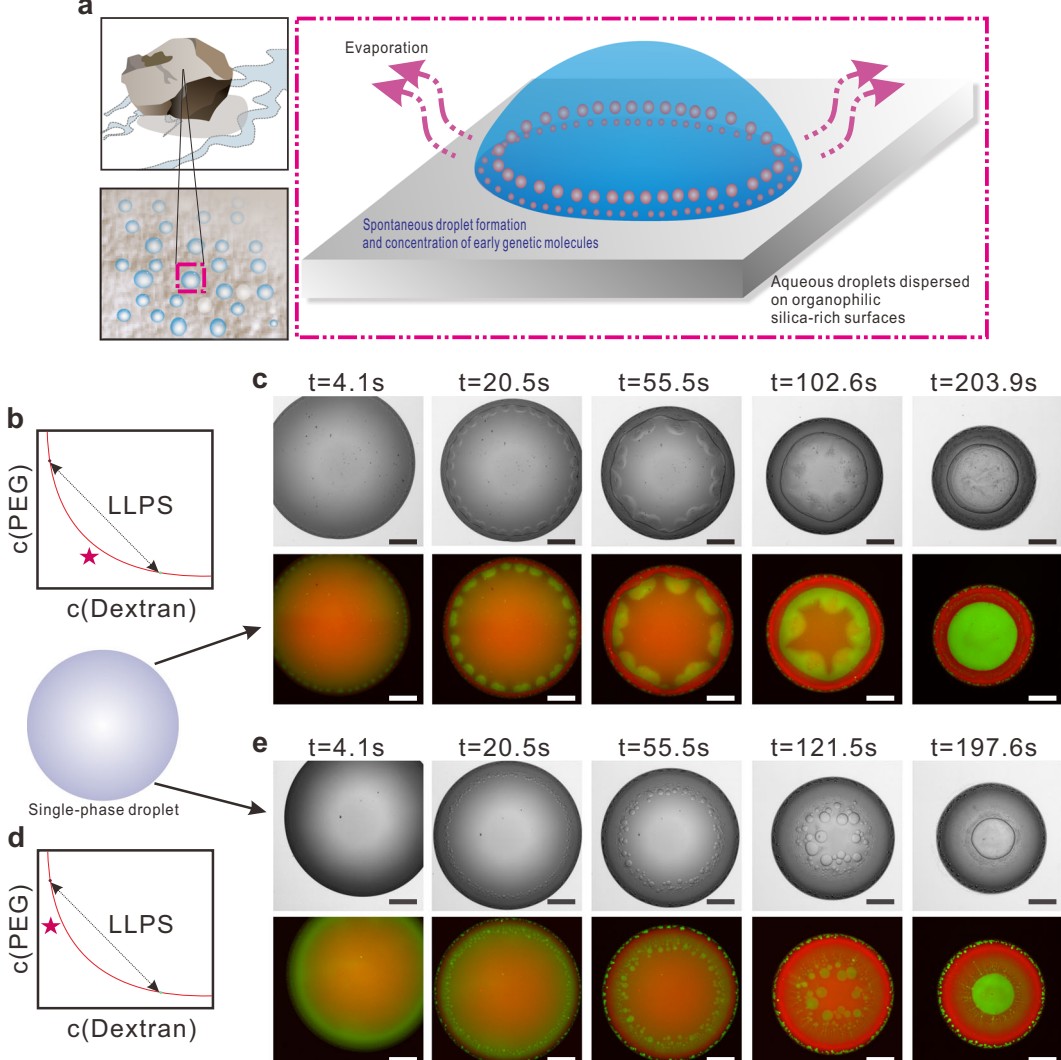

**Fig. 1 Evaporation-triggered segregative LLPS inside the all-aqueous sessile droplet. a** Schematic drawing of early genetic molecule compartmentalization inside the evaporating aqueous droplets. According to timeline of the early history of life, the Pre-RNA world and RNA world may have occurred at 3.8–4.0 Gya[77], with the first RNA polymers formed in warm little ponds[78]. Silica-rich surface could serve as the reactor that supports various prebiotic reactions such as the polymerization of the polynucleotides;[42,79] **b, d** Phase diagrams of PEG and dextran mixtures. The red solid line is the binodal curve that distinguishes the single-phase region and the two-phase coexistence region. The arrowed line represents the tie line of ATPS, along which a mixture undergoes liquid-liquid phase separation (LLPS) and forms a PEG-rich phase and a dextran-rich phase. The star represents the composition of the sessile droplet, with 5 wt% PEG–10 wt% dextran in (**b**) and 9 wt% PEG–4 wt% dextran in (**d**); **c** Phase-separated pattern evolution inside an evaporating droplet of regime 1, shown in both bright-field image sequence and fluorescence image sequence, respectively; The dextran-rich phase is labeled by fluorescein isothiocyanate-dextran (FITC-dextran, green) and PEG-rich phase is labeled by Rhodamine B (red). **e** Phase-separated pattern evolution inside an evaporating droplet of regime 2, shown in bright-field image sequence and fluorescence image sequence, respectively; The fluorescence labeling is the same with that in (**c**). For **c** and **e**, images are obtained over analysis of seven independent trials with relative humidity ranging from 55 to 65%. The scale bar is 500 μm.

green fluorescence profile inside the droplet. For a sessile droplet consisting of 5 wt% PEG and 10 wt% dextran (Regime 1, Fig. 1b), a lobe-shaped dextran-rich phase forms shortly after the evaporation begins (Fig. 1c and Supplementary Movie 1). For the droplet consisting of 9 wt% PEG and 4 wt% dextran (Regime 2, Fig. 1d), there is no such lobe-shaped area; instead hundreds of dispersed dextran-rich droplets are formed (Fig. 1d and Supplementary Movie 2), mainly due to the relatively lower initial concentration of dextran in this regime.

Despite the distinct main LLPS patterns observed, the two regimes share many similar characteristics during the entire droplet evaporation process. Just at the beginning of evaporation, the sessile droplet is well mixed with a single-phase composition,

as revealed by the uniform single color at the bottom. After a few seconds, small dextran-rich compartments with green fluorescence start to nucleate at the rim of the droplet. For both regimes, the nucleation of dextran-rich compartments near the three-phase contact line is continuously triggered once the polymer concentration surpasses the binodal curve (Supplementary Fig. 1 and Supplementary Note 1). In regime 1, the dextran-rich compartments coalesce quickly after nucleation, forming lobe-shaped domains of dextran-rich phase (Fig. 1c). In regime 2, these small compartments remain as dispersed droplets (Fig. 1d). During further evaporation, these nucleated dextran-rich compartments, whether in the form of the lobe-shaped domains or small droplets, keep coarsening and growing, while they are

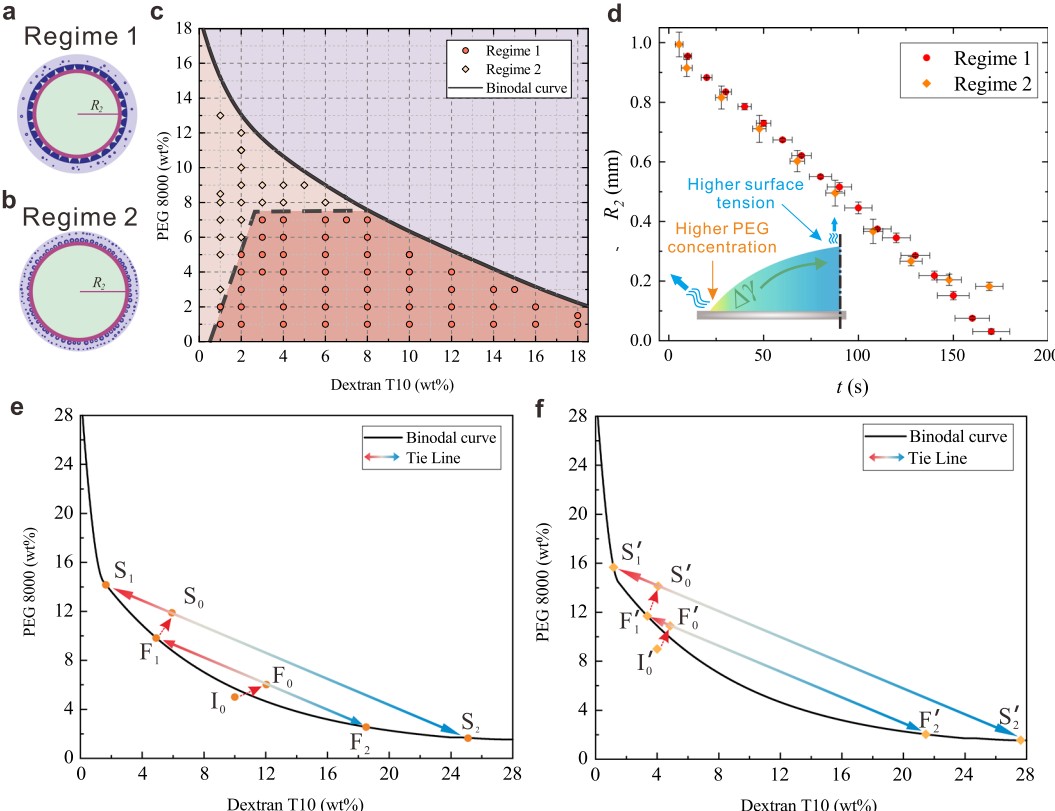

**Fig. 2 Phase separation dynamics inside the evaporating droplet. a**, **b** Schematic drawings for polymer self-organized patterns in regime 1 (**a**) and regime 2 (**b**), with the definition of LLPS front that distinct single-phase region (light green color) and phase separated region (light purple for PEG-rich phase and dark purple for dextran-rich phase) inside the sessile droplet, which are demonstrated by the red solid circles with a radius of $R_2$; **c** Phase diagram of evaporation-triggered polymer self-organization regimes inside the sessile droplet; The phase diagram is the result of observations and analysis of at least three independent trials for each data point, with relative humidity ranging from 55 to 65%. **d** The radius of LLPS front $R_2$ as a function of evaporation time. Error bars represent SEM (standard error of the mean) from three independent experiments; Insert: Schematic drawing of nonuniform evaporation rate induced Marangoni effects. A surface tension difference $\Delta\gamma$ is generated due to the nonuniform evaporation flux along the droplet surface. **e**, **f** Kinetic pathway of LLPS in regime 1 (**e**) and regime 2 (**f**). A finite small area ($\triangle r = 0.05R_0$) near droplet edge is chosen as the calculation domain, with a time step (denoted by red arrows) of $\triangle t = 5s$; $I_0$ and $I_0'$ are initial compositions of the single-phase droplet. $F_0$ and $F_0'$ are compositions within the calculation domain after the first time step, with their exact values given in Supplementary Table 1. Once surpassing the binodal curve, the mixture spontaneously phase separates into a PEG-rich phase ($F_1$ and $F_1'$) and a dextran-rich phase ($F_2$ and $F_2'$), respectively. Similarly, $S_0$ and $S_0'$ correspond to the compositions within the calculation domain after the second time step (Supplementary Table 1), with the formation of a new PEG-rich phase ($S_1$ and $S_1'$) and a new dextran-rich phase ($S_2$ and $S_2'$), respectively. Source data are provided as a Source Data file.

hydrodynamically advected to the middle of the sessile droplet. Eventually, the dextran-rich phase covers the inner part of the sessile droplet until the evaporation process stops.

In our model system, both dextran-rich and PEG-rich domains nucleate quickly at the rim of the droplet due to phase separation. However, since PEG is more hydrophobic than dextran[45] and can adsorb onto silica[46], the dextran-rich compartments, though with a higher density than the PEG-rich phase[37], are advected to the inner part by the fluid flow and PEG-rich compartments stay outside the sessile droplet, as suggested by the SEM images of the deposits (Supplementary Fig. 2).

**Kinetic pathway of nonassociative phase separation under the NES**. As can be seen from the experimental results, under the NES of droplet evaporation, polymer concentrations play an important role in determining LLPS patterns and their evolution (Fig. 1b, d). We then test sessile droplets containing ATPS mixtures with different concentrations of PEG and dextran. In all the test, polymers are dissolved with a concentration below the binodal curve so that they are homogeneously in a single phase before evaporation. The phase diagram of evaporation induced LLPS patterns for these mixtures is then obtained (Fig. 2c). In

general, for droplets with the concentration of dextran lower than 4 wt%, evaporation triggers dispersed LLPS patterns (regime 2) when the concentration of PEG is more than two times higher than that of dextran. For droplets with dextran concentration higher than 4 wt%, there is a critical concentration of PEG (about 7 wt%) to determine LLPS regimes. We can observe dispersed LLPS patterns (regime 2) when the concentration of PEG is higher than this value. Otherwise, lobe-shaped LLPS patterns (regime 1) will appear. More importantly, as shown in the phase diagram of LLPS patterns (Fig. 2c), polymer self-organization can be triggered by water evaporation with very little amount of PEG and dextran additives (about 1 wt%), thus providing a robust strategy for prebiotic compartmentalization.

The phase diagram indicates that initial polymer concentration is an important factor that controls LLPS patterns (Fig. 2a–c). As a first qualitative understanding, water loss due to evaporation will induce the increase of polymer concentrations inside the droplet, which enhances nonspecific molecular interactions and triggers polymer incompatibility. As a result, phase-separated domains are nucleated inside the droplets and coalesce into PEG-/dextran-rich phase-separated regions. The phase-separated regions start to appear at the rim of the sessile droplet due to the

higher local evaporation rate, and they evolve with further water loss to form the LLPS patterns.

To obtain a quantitative analysis of the nonassociative phase separation triggered by evaporation, we calculate the evaporation rate of the droplet to obtain the change of polymer concentrations, which determines kinetic pathway of phase separation. Evaporation induced phase segregation inside the evaporating sessile droplet of a binary and ternary mixture, with organic components involved, has been reported recently[47,48]. For a sessile droplet of pure liquid, its evaporation process is dominated by vapor diffusion from the liquid-air interface to the surroundings. The analytical model of pure sessile droplet evaporation has been solved and shown as an accurate description of the evaporation dynamics[49]. In our model system, despite the complex LLPS phenomena inside the evaporating droplet, the evaporation process is indeed still governed by the vapor diffusion from droplet surface to surroundings[47], so we adopt the evaporation model for a pure sessile droplet to describe the water loss process in our experiments. The mass change rate of an evaporating water droplet depends on the droplet radius $R$, vapor diffusion coefficient $D$, saturated vapor concentration $n_s$, and ambient vapor concentration $n_\infty$, given by

$$\frac{dM}{dt} = -\pi R D (n_s - n_\infty) f(\theta) \tag{1}$$

where

$$f(\theta) = \frac{\sin\theta}{1 + \cos\theta} + 4 \int_0^\infty \frac{1 + \cosh 2\theta\tau}{\sinh 2\pi\tau} \tanh[(\pi - \theta)\tau] d\tau \tag{2}$$

is the function of droplet contact angle $\theta$. To test this assumption, we measure the evaporation rate of a pure water droplet and a droplet of the aqueous solution containing 10 wt% PEG and 5 wt % dextran (Supplementary Figs. 3 and 4). The rate of mass change of both sessile droplets shows very good agreement and is consistent with the theoretical model, indicating that our assumption is reasonable. Hence, we can estimate the nonuniform evaporation rate and flux of the droplets in our experiments based on the evaporation model of a sessile droplet of pure water.

LLPS begins at the rim of the sessile droplet because of the much higher evaporation flux here than that in the middle part of the droplet, as suggested by the evaporation flux distribution $J(r) \propto (R_0 - r)^{-0.5}$, where $R_0$ is the initial droplet radius and $r$ is the horizontal distance to center of the sessile droplet[49]. Due to the higher evaporation flux near the rim of the droplet (Supplementary Fig. 5), concentrations of PEG and dextran rapidly increase and cross over the binodal curve quickly, triggering LLPS and formation of phase-separated compartments. To simplify the problem, we select a finite small area ($\triangle r = 0.05R_0$) near droplet edge as the calculation domain and estimate the increase in the polymer concentration within this domain. For both LLPS regimes demonstrated in Fig. 1, phase-separated compartments are clearly visible within a few seconds. Hence, to supplement the experimental findings and generalize the understanding of LLPS, we estimate the kinetic pathway based on the polymer concentration increase in the calculation domain with a time step ($\triangle t = 5$ s). The detailed calculation is given in the Supplementary Method 1.

The calculated local polymer concentration increase induced by evaporation (Supplementary Table 1), and the corresponding kinetic pathways of the two regimes are shown in Fig. 2e, f, respectively. Initially, the single-phase mixtures are homogeneous without any phase-separated compartments ($I_0$ and $I_0'$). After 5 s of evaporation, the polymer concentrations increase and surpass the binodal curve, triggering LLPS inside the calculation domains ($F_0$ and $F_0'$). The composition and mass fraction of separated

phases, including PEG-rich ($F_1$ and $F_1'$) and dextran-rich ($F_2$ and $F_2'$), are determined by the tie line in the phase diagram. For LLPS pattern evolutions in regime 1, the ratio of mass fraction of dextran-rich phase to that of the PEG-rich phase equals to the length ratio $\overline{F_0F_1}/\overline{F_0F_2}$ ($\approx 1.1$), suggesting that the dextran-rich phase and the PEG-rich phase formed by the first LLPS in the calculation domain have similar mass fractions. This allows the phase-separated dextran-rich compartments to coalesce quickly and form the continuous lobe-shaped structures. For LLPS pattern evolutions in regime 2, the ratio of mass fraction of dextran-rich phase to that of PEG-rich phase equals to the length ratio $\overline{F_0'F_1'}/\overline{F_0'F_2'}$ ($\approx 0.09$), which means that the dextran-rich phase has a mass fraction over ten times less than that of the PEG-rich phase. The low mass fraction of the dextran-rich phase inhibits the formation of the continuous lobe-shaped structures, but leads to the formation of dispersed droplets instead. Further evaporation of the water solvent in the calculation domain enhances the LLPS in the same manner ($S_i$ and $S_i'$, $i = 0, 1, 2.$). From the kinetic pathway of phase separation, we conclude that, besides the initial composition of the mixture, the compositional change due to water loss also affects the dynamics of LLPS in the sessile droplet.

**Marangoni stresses induced by the nonuniform evaporation flux.** From pattern evolutions of LLPS in both regimes, we observe that the phase-separated dextran-rich compartments always move towards the center of the sessile droplet, rather than being advected to droplet edge by outwards capillary flows, as in systems exhibiting the so-called "coffee-ring effects"[50]. To quantitatively understand the dynamics of phase-separated compartments, we measure the displacement of the phase separation front (PSF) inside the evaporating droplet, which is defined as the front of the phase-separated regions inside the sessile droplets (Fig. 2a, b). Figure 2d shows the radius of PSF, $R_2$, as a function of evaporation time. For both LLPS regimes, $R_2$ linearly decreases with the evaporation time, which means that the displacement of PSF, $L = R - R_2$, has a scaling of $L \sim t$. This is different from the typical diffusion-dominated process, where the movement of different phases is driven by the concentration gradient and $L \sim t^{1/2}$ is expected[51,52].

We attribute this reverse "coffee-ring effects" to the Marangoni flow caused by the surface tension gradient along the surface of the sessile droplet, as shown in the insert of Fig. 2d. Our assumption is based on three facts: (i) during the evaporation process, the surface tension force is dominant when compared to gravitational forces, as the Bond number Bo$= \rho g R^2/\gamma \approx 0.1$, where $g = 9.8$ m s$^{-2}$ is gravity and $\gamma = 64$ m N m$^{-1}$ is the surface tension of the sessile droplet; (ii) the velocity of PSF is almost constant and calculated to be about 6 μm s$^{-1}$, in the same scale as that of Marangoni-driven fronts of thin spreading films;[53] (iii) as a water-soluble polymer, PEG is a surface-active agent capable of decreasing surface tension of the solution (Supplementary Fig. 6), while an enlarged compositional gradient inside the droplet appears immediately upon LLPS, just like the much more compositional differences between $F_2$ (or $F_1$) and $I_0$ than that between $F_0$ and $I_0$.

To estimate the Marangoni effects, we use the lubrication approximation[54] to simplify the model system. By balancing the capillary pressure and Marangoni stress (Supplementary Method 2), an estimated surface tension difference $\Delta\gamma \approx 2$ m N m$^{-1}$ is sufficient to induce such an inward Marangoni flow that is comparable to the outward capillary flow. Considering that in our model system, $\Delta\gamma \approx 3.5$ m N m$^{-1}$ can be easily achieved due to the highly nonuniform evaporation flux along the surface of the sessile droplet (Supplementary Fig. 6), we can conclude that in

both LLPS regimes, the Marangoni flow driven by the surface tension differences plays a dominant role in the movement of phase-separated compartments inside the sessile droplet.

The evaporation of the sessile droplet allows us to further study the formation of the phase-separated compartments from the microscopic perspective. The coarsening process of these compartments is likely due to transport by the hydrodynamic flow (Supplementary Fig. 7a and Supplementary Note 2). As shown by the SEM and confocal imaging (Supplementary Fig. 7c, d), the diameter of the small nucleated droplets ranges from hundreds of nanometers to several microns, corresponding to a volume on the order of attoliters. Droplet "nuclei" with a smaller size than submicron-sized dextran-rich droplets in Supplementary Fig. 7e could be expected at the initial stage of polymer phase separation[55]. The size of these compartments formed through LLPS is comparable to that of phase-separated organelles in the living cells. It is difficult to make such small sized ATPS droplets by traditional droplet microfluidics techniques[56]. This provides great potential for various droplet-based bioinspired techniques in our model system[57–60]. The "pinning" of the dextran-rich droplets near the rim is observed before the drying of the sessile droplet (Fig. 1c, e), which is attributed to the reduced droplet height and thus the confined space for the inhibition of inward Marangoni convection (Supplementary Fig. 1e, f).

Summarizing, our results present a robust NES for nonassociative phase separation inside a sessile droplet. Due to the highly nonuniform evaporation flux, phase separation is triggered at the rim of the sessile droplet, exhibiting a self-organized pattern decided by both original composition and the concentration change due to the water loss, as demonstrated by the kinetic pathway of phase separation. Meanwhile, the emergence of phase separation can enlarge the compositional gradient inside the droplet, triggering Marangoni flow that convects the phase-separated compartments to the center of the droplet. Hence, the microstructures of phase separation and the hydrodynamic flow are strongly coupled inside the evaporating droplet. With the appearance of rich dynamics, a full understanding on the phase behavior and pattern evolution during droplet evaporation would require the coupling of the Navier–Stokes equation and Cahn–Hilliard equation[61,62]. However, the resulting high nonlinearity suggests that it could only be solved by full direct numerical simulations. Nevertheless, kinetic pathways of LLPS obtained by spatiotemporal averaging of polymer concentrations due to water loss have revealed the increase of the tie line length (TLL) due to evaporation. In the PEG/dextran system, an increase of TLL can lead to the increase of the density difference, the interfacial tension, the viscosity of the dextran-rich phase, as well as partitioning strengths of a third species like nucleic acids and proteins into one of the two phases[63]. Hence, through understanding the kinetic pathway and increasing TTL, our model system can be further harnessed to concentrate biopolymers inside phase-separated compartments, and to serve as the reactor that can enhance a series of biochemical reactions.

**Encapsulation of informational molecules through membraneless compartmentalization**. Here we point out some common features of polymer self-organization and compartmentalization in our model system with that of the intracellular LLPS:[15] (i) They all undergo a transition from a homogeneous state to a phase-separated state, and that only happens above a threshold concentration of the components; (ii) they are both surrounded by the aqueous macromolecular-crowding environment and governed by nonequilibrium thermodynamics, and (iii) the immiscible compartments formed by LLPS both exhibit properties, such

as round morphology, fusion by contact, and wetting properties, that characterize typical liquid structures. These similarities make it possible to bridge prebiotic LLPS to protocells with subcellular membraneless compartments by using our model system.

The RNA world is a leading hypothesis for the origins of life[64]. As RNA is a polymer capable of serving as both information storage and a functional catalyst, it is hypothesized to be intrinsic to the origins of life. RNA can undergo associative LLPS via complex coacervation in the presence of polycations, as it has a polyanionic backbone. Likewise, RNA building blocks nucleoside triphosphates (NTPs) can undergo LLPS along with polyphosphate, which is hypothesized to be a primordial energy molecule[65]. Associative LLPS has previously been shown capable of concentrating such biomolecules, however the structural stability and functionality of highly-charged biopolymers like DNA and RNA are inhibited due to their complex coacervation with polycations[25,28]. Thus a nonassociative phase separation model with plausible prebiotic conditions is required, where biopolymers can be concentrated without the loss of stability and functionality.

To explore the role of nonassociative phase separation in biopolymer self-organization and compartmentalization under a prebiotic environment, we firstly introduce DNA and RNA separately into our model system. To monitor DNA localization, a Cy-5 labeled (red fluorescence) ssDNA sequence was used in combination with FITC-dextran (green fluorescence) to label the dextran-rich phase. The DNA localized to the dextran-rich phase by LLPS (Fig. 3 and Supplementary Movie 3). This indicates that, with the functions of polymer self-organization and compartmentalization, our model system can provide a prebiotic strategy for enrichment and encapsulation of biopolymers. Similarly, RNA localized in the dextran-rich phase when introducing the fluorescent Broccoli RNA aptamer/DFHBI-T1 pair into the model system (Supplementary Fig. 8 and Supplementary Movie 4). This is consistent with previous findings that hydrophilic components like DNA and RNA partition into dextran-rich phase[27,36]. For both the Cy-5 labeled DNA and the Broccoli RNA aptamer/DFHBI-1T, the ratio of the fluorescence intensity of dextran-rich compartments ($I_d$) to that of continuous phase ($I_c$), $K = I_d/I_c$, has a greater than two-order-of-magnitude increase after $\sim 4$ min of evaporation, indicating effective enrichment through compartmentalization. The increase of $K$ indicates an increase of partitioning strength of DNA and RNA inside the dextran-rich compartments during evaporation. This is consistent with an increasing TLL determined from the kinetic pathways of phase separation (Fig. 2e, f). Localization and compartmentalization of DNA and RNA inside the evaporating droplet of regime 1 are given in Supplementary Fig. 9a, b, respectively. The compartmentalization of DNA and RNA provides a strategy to inhibit their thermal diffusion, thereby supporting chemical reactions in the dilute primordial soup. This spontaneous compartmentalization and concentration show a decrease in the entropic state of DNA and RNA.

These DNA/RNA-containing dextran-rich liquid droplets can range from hundreds of nanometers to several microns. Similar to intracellular membraneless compartments composed of RNA and proteins, the DNA/RNA-containing dextran-rich liquid droplets have interfacial tension-directed movement and coarsening behaviors. This implies that polymer self-organization and compartmentalization could have played a significant role in driving the evolutionary engine of the first live cells under prebiotic conditions.

To demonstrate the advantages of maintaining functionality of nucleic acids through nonassociative phase separation, we perform IVT inside our model system (Fig. 4 and Supplementary

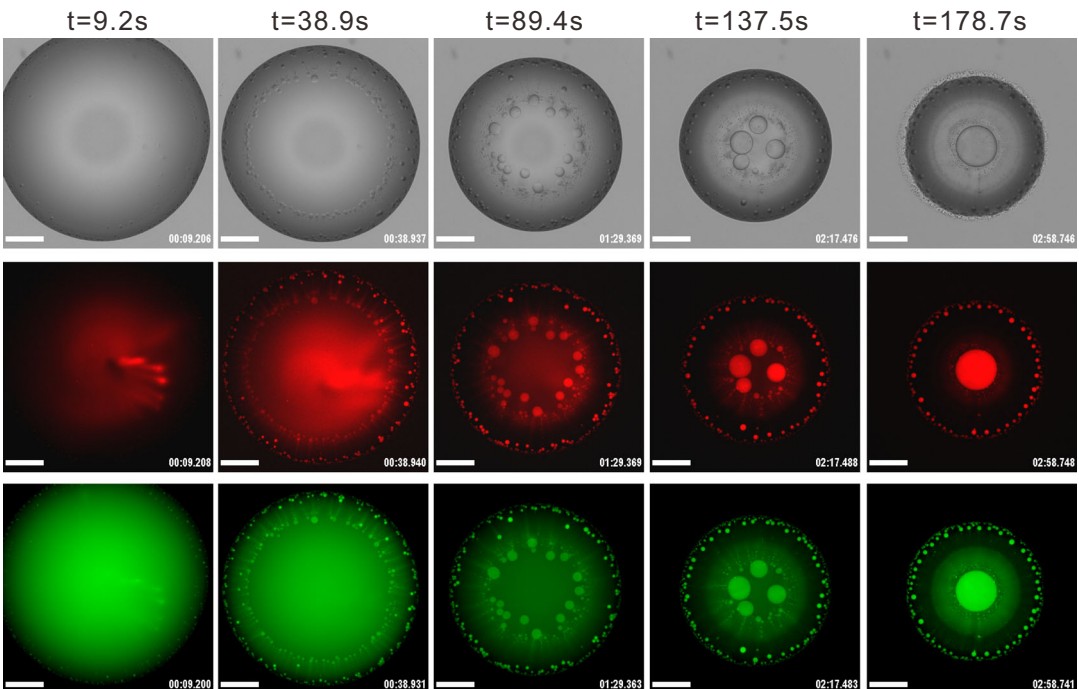

**Fig. 3 Compartmentalization and localization of DNA inside the evaporating droplet.** DNA localizes into dextran-rich compartments inside the evaporating sessile droplet. DNA is labeled with Cy-5 (red) and dextran-rich compartments are labeled with FITC-dextran (green). The ATPS solution is made up of 9 wt% PEG and 4 wt% dextran, with the addition of Cy-5 labeled DNA (10 μM) and FITC-dextran ($M_w$ = 4000, 0.5 wt%). Images are obtained over analysis of seven independent trials with relative humidity ranging from 55 to 65%. The scale bar is 500 μm.

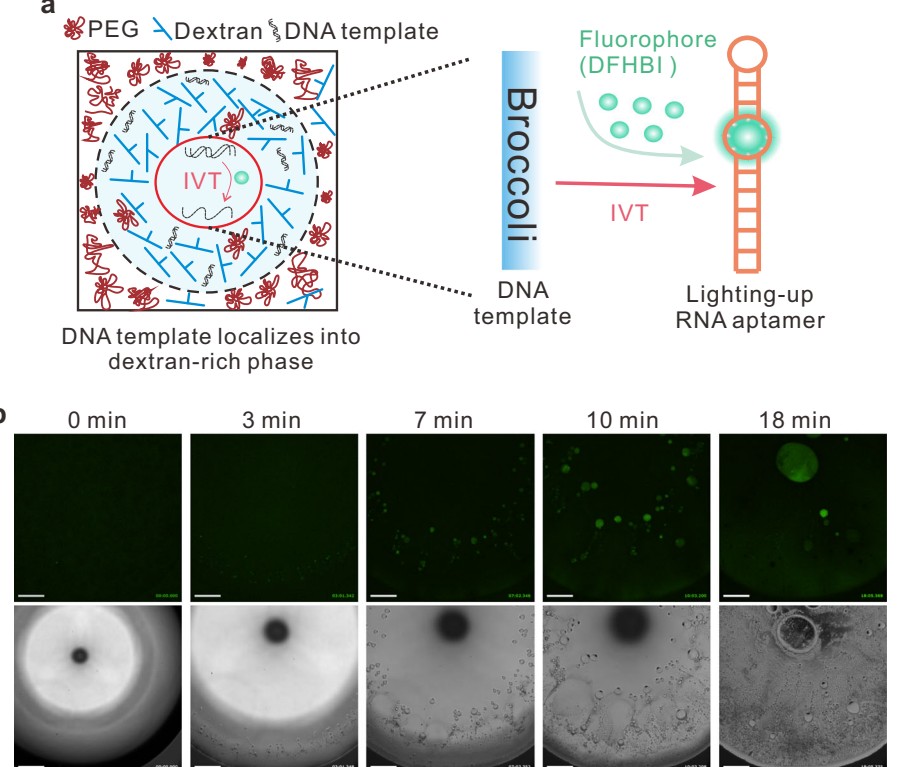

**Fig. 4 In vitro transcription of functional RNA aptamers inside the evaporating sessile droplets. a** The flow of genetic information in phase-separated compartments via in vitro transcription (IVT). The DNA templates are coded for Broccoli aptamer that can bind 5-difluoro-4-hydroxybenzylidene imidazolinone (DFHBI) to form a fluorescent complex of Broccoli–DFHBI. **b** Optical image sequence shows fluorescence RNA aptamer transcription from Broccoli DNA template inside the phase-separated compartments. The dextran-rich compartments have increasing green fluorescence due to the formation of Broccoli-DFHBI complexes. Images are obtained over analysis of three independent trials with relative humidity ranging from 55 to 65%. The scale bar is 500 μm.

Movie 5). The sequence of confocal images shows high fluorescence in the dextran-rich compartments due to RNA synthesis, while a lower signal was observed in the surrounding PEG-rich phase. As the DNA partitions into the dextran-rich phase, there is no clear fluorescence increase outside the compartments. Therefore, we conclude that transcription occurs exclusively in the phase-separated compartments. A similar transcription reaction inside phase-separated dextran-rich compartments within the evaporating droplet of regime 1 is shown in Supplementary Fig. 9c. To confirm that transcription does occur inside the phase-separated dextran-rich droplets, we added Cy-5 labeled Broccoli DNA template into the ATPS droplets and observed the localization of both DNA template and transcribed RNA aptamers. The results show that DNA localizes into dextran-rich droplets during evaporation and the fluorescence of RNA-fluorophore complexes can be only observed inside the dextran-rich compartments (Supplementary Fig. 10). Hence, our model system can provide a cell-mimicking environment that supports the flow of genetic information. The compartmentalization of functional biopolymers provides a plausible strategy to assemble molecules relevant to the RNA world hypothesis. Under similar prebiotic compartmentalization conditions, such a mechanism could have played a role in the evolution of informational polymers on the early Earth.

**Enhanced ribozyme cleavage by compartmentalization**. According to the RNA world hypothesis, self-replicating RNA molecules underwent evolution leading to the emergence of life[65,66]. Various functions of catalytic RNA under different prebiotic conditions have been demonstrated. In particular, it has been suggested that water evaporation could have led to a viscous environment under model prebiotic conditions, thereby enhancing RNA replication and catalytic activities[67,68]. We suggest that our model system could provide a prebiotic strategy for the concentration of catalytic RNAs, thus promoting their functionality. To demonstrate this argument, we performed an RNA substrate cleavage reaction using hammerhead ribozyme (HHR)[69] in the evaporating ATPS droplet. The substrate (HHS) is labeled terminally with a FAM fluorophore and a TAMRA FRET pair quencher. Under catalysis by HHR, the substrate is cleaved into two fragments and emits fluorescence (Fig. 5a, Supplementary Movie 6, and Supplementary Fig. 11). This fluorescence activation substrate system was used to visualize the ribozyme cleavage reaction in real time. Figure 5b shows the image sequence of an evaporating sessile droplet with a volume of 5 μL after HHS and HHE are introduced. We can observe that catalysis occurring in fluorescent dextran-rich compartments is moving towards the center of the sessile droplet during evaporation, showing increased compartmentalization. To further prove that RNA catalysis occurs during the evaporation process, two control groups were used, "no ribozyme" and "no ATPS". For the "no ribozyme" control group where there is only HHS added without any HHR (Supplementary Fig. 12), we can also observe fluorescent domains in the evaporating droplet (Fig. 5c) albeit with a lower relative fluorescence level. The pattern formed in this control group differs from the experimental group in that most of the fluorescent dextran-rich regions stay at the rim of the droplet (Fig. 5c). For the "no ATPS" control group, both the fluorophore-labeled substrate and enzyme strand were introduced into the solution without the reagents required for ATPS (Supplementary Fig. 13). The results of "no ATPS" control group present much lower ribozyme cleavage activity (Supplementary Figs. 13 and 14), indicating the unique contributions of compartmentalization via nonassociative phase separation in prebiotic chemistry.

To quantify the fluorescence levels of sessile droplets over time, we take the flattened y-axis intensity signal within the dashed red rectangle (Fig. 5b, c). This provides the spatial evolution over time of the fluorescent areas in the evaporating droplet, as shown in Fig. 5d. We can see that for the experimental group, the intensity is increasing over time, with the peak of the signal moving towards the center. This is in contrast to the "no ribozyme" control group in which the intensity is much lower and the fluorescence signal localizes at the edge of the sessile droplets. Additionally, the no ATPS control group shows no compartmentalization and lower fluorescence (Fig. 5d). These distinct differences not only suggest that the ribozyme cleavage reaction proceeds successfully in our model system, but that the reaction rate is increased by over three times (Supplementary Fig. 14). This increase in reaction rate was confirmed with wet-dry cycling experiments analysed using a gel electrophoresis-based analysis method (Supplementary Fig. 15). Similarly, ribozyme cleavage can also occur inside phase-separated dextran-rich compartments within the evaporating droplet of regime 1 (Supplementary Fig. 9d). Taken together with the advantages in scalability of our model system, we conclude that compartmentalization through nonassociative LLPS would have significant impact on acceleration of RNA self-replication and evolution under prebiotic conditions.

To illustrate the universal plausibility of our model system to support RNA catalysis reactions, we tested another RNA ribozyme, X-motif[66], inside the evaporating sessile droplet (Supplementary Fig. 16). A similar cleavage reaction is achieved by evaporation induced nonassociative phase separation and compartmentalization, identifying the advantages of our model system in maintaining the stability and enhancing the functionality of catalytic biopolymers.

To gain insights on the mechanism of the enhanced ribozyme cleavage by segregative phase separation, we compared the reaction kinetics inside the evaporating ATPS droplet and the water droplet. The kinetics of the HHR cleavage has been well characterized, and follows the Michaelis–Menten scheme $S \overset{E}{\leftrightarrow} S \cdot E \overset{E}{\to} P$[70], where $S$ is the substrate, $E$ is the ribozyme, $S \cdot E$ is the substrate-ribozyme complex, and $P$ is the product. In the case of HHR cleavage, $P$ should actually be replaced by two shorter RNA chains ($P_1$ and $P_2$). Here we assume the complexing of substrate and ribozyme, and the cleavage of the complex to be realized immediately[71]. The concentration of $S \cdot E$ is accordingly considered to be negligibly low. Hence, we model ribozyme cleavage as the simplest generic case when the reaction runs via the step of $S \overset{E}{\to} P$. The partitioning of oligonucleotides in ATPS is length dependent. In our model system, the chain of ribozyme (43 nt) is much longer than the chains of product (8 nt and 6 nt), leading to a strong partitioning for the ribozyme and much weaker partitioning for the products. This partitioning difference results in a phase-separated dextran-rich droplet reactor that concentrates long ribozyme chains but is permeable for short product chains. Therefore, this drives the reaction inside the compartment towards the cleavage products[36].

We developed a simplified theoretical model by considering reactions inside a small domain with a radius of 10 μm (as shown in the schematic in the Supplementary Method 3). Specifically, inside the ATPS droplet, a domain of a phase-separated dextran-rich compartment surrounded by PEG-rich phase is chosen, while inside the water droplet, the small domain has the same component with the surrounding fluid. Within the domain, $S$ and $P$ are assumed to move by diffusion with equal diffusion constants, while the ribozyme uniformly distributed throughout the small domain. This assumption is consistent with many cellular reactions where diffusion of $E$ is neglected, because the molecular weight of ribozymes is much larger than those of the reactants. The concentrations of $S$ and $P$ inside the domain, $s(x, t)$

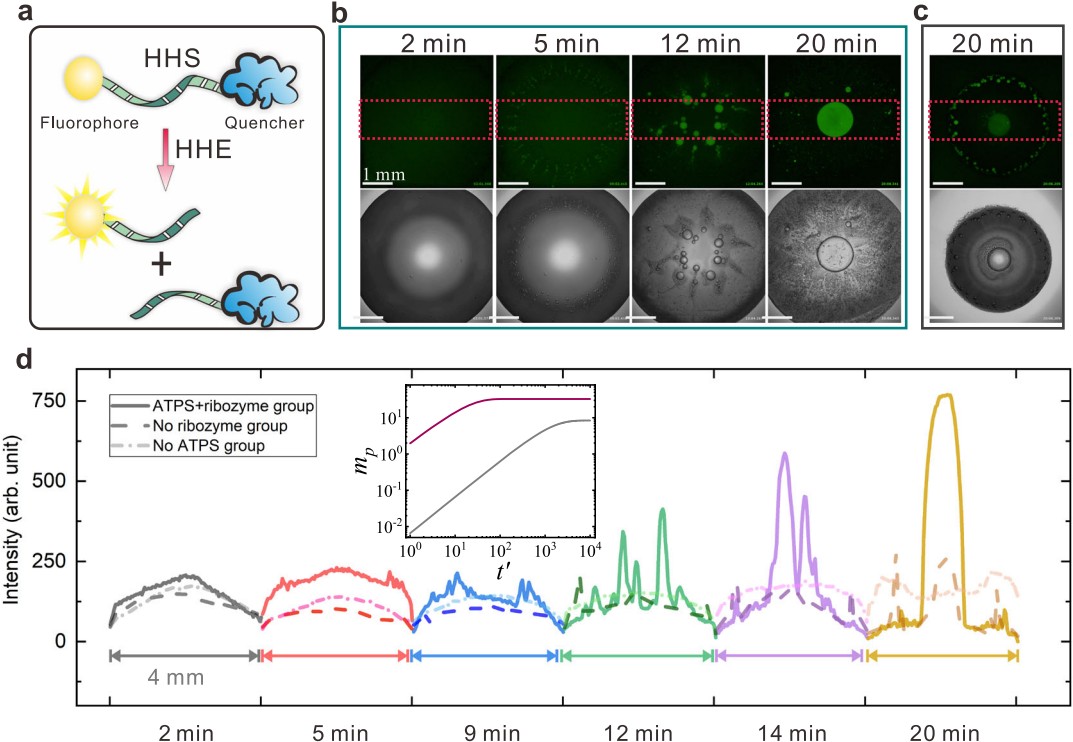

**Fig. 5 Enhanced ribozyme cleavage by compartmentalization. a** Schematic drawing of the RNA cleavage reaction. The fluorophore labeled hammerhead substrate (HHS) is cut into two smaller pieces by the hammerhead ribozyme (HHE) and emits green fluorescence. The substrate contains a donor that is quenched by FRET when HHS is not cleavage. **b** Image sequence shows RNA cleavage process inside the phase-separated compartments. **c** Control group of the RNA cleavage reaction inside an evaporating droplet while without HHE added. There is only fluorophore labeled substrate (HHS) introduced into the droplets. **d** Fluorescence intensity evolution of experimental group (both HHS and HHE are introduced into the evaporating ATPS droplet; Solid line), no ribozyme control group (only HHE is introduced into the evaporating ATPS droplet; Dashed line) as well as no ATPS control group (both HHS and HHE are introduced into the evaporating water droplet; Dash-dotted line). The intensity value is obtained by gel analysis of a specified rectangular area (as shown in (**b**) and (**c**)) in the fluorescence images. Images are obtained over analysis of three independent trials with relative humidity ranging from 55 to 65%. Insert: Dimensionless productivities $m_p$ of ribozyme cleavage in the domain of dextran-rich compartment (red solid line) and that of water droplet (gray solid line) as a function of dimensionless time $t'$. Source data are provided as a Source Data file. The scale bar is 1 mm.

and $p(x, t)$, respectively, therefore follow the coupled reaction-diffusion equations[72]

$$\frac{\partial s(x, t)}{\partial t} = D\nabla^2 s(x, t) - \frac{k_{cat} e s(x, t)}{K_M + s(x, t)} \tag{3}$$

$$\frac{\partial p(x, t)}{\partial t} = D\nabla^2 p(x, t) + \frac{k_{cat} e s(x, t)}{K_M + s(x, t)} \tag{4}$$

where $D$ is the RNA diffusion constant inside the domain and $e$ is the concentration of ribozyme. $k_{cat}$ and $K_M$ denote catalytic rate and Michaelis constant, respectively. The last term in the first equation represents the conversion of $S$ into $P$ by $E$, which appears as a local source of $P$ in the second equation. Details of the reaction kinetic modeling inside the dextran-rich compartments and water droplets are given in supporting information (Supplementary Method 3). The effects of length-dependent RNA partitioning in the domain of dextran-rich compartment are modeled via setting different boundary conditions (Supplementary Fig. 17).

Our model does not consider complex interactions between the ribozyme cleavage reaction and the phase separation dynamics inside the evaporating droplet, such as mass transport and content exchange due to droplet nucleation, movement, and coalescence. These factors would be both time- and composition-dependent and beyond the scope of this paper. The model (Eq. 3, Eq. 4, and Supplementary Method 3) provides essential insights

by introducing RNA partitioning effects and setting different boundary conditions related to the segregative phase separation.

The resulting minimal model is shown to capture the essential results from experimental observation. With the length-dependent RNA partitioning, the distribution of substrates and products inside the dextran-rich compartments is significantly different from that of the water droplet (Supplementary Fig. 18a–f). The insert of Fig. 5d shows the productivity comparison of the ribozyme cleavage reaction inside the domain of dextran-rich compartment and that of water droplet, where it can be found: (1) the time required to reach the maximum productivity in dextran-rich compartments is about 50 times less than that in water droplets; (2) the final reaction productivity in the dextran-rich domain is about four times higher than that in water droplets. Therefore, we conclude from the reaction-diffusion model that due to the partitioning and compartmentalization effects, RNA catalysis reactions can be largely enhanced inside the dextran-rich droplets. The model represents an ideal case where (1) only a stationary phase-separated dextran-rich domain in the evaporating ATPS droplet is considered; (2) the supply of substrate to the dextran-rich phase is always constant; and (3) no loss from degradation or RNAse is included, and thus there could be an overestimation of the reaction rate inside the dextran-rich compartments. However, the results from the model are qualitatively consistent with our experimental observations that the fluorescence signal appears much faster and reaches a higher intensity in evaporating ATPS droplets than in water

droplet. Hence, we can conclude that the dextran-rich compartments formed through segregative phase separation have significant advantages over water droplets in both accelerating and promoting ribozyme cleavage, which arise from a spatial localization and enrichment of the reactant RNA inside the dextran-rich compartments triggered by phase separation. More importantly, our results show that RNA can be compartmentalized based on its length via segregative phase separation of ATPS. Longer sequences are concentrated into the compartments while shorter pieces are released. Such length-dependent RNA compartmentalization provides plausibility that phase-separated compartmentalization could have served as a primitive sorter during early molecular evolution.

## Discussion

The RNA world hypothesis is a prevailing model for origin of life research, where RNA acted both as a catalyst and as a genetic storage molecule. In the RNA world, prebiotic compartmentalization would have been essential to enrich nucleic acids from the dilute primordial environment to a sufficient concentration for reactions and molecular evolution. Here, we have presented a model system for prebiotic compartmentalization strategies by harnessing segregative LLPS within an evaporating sessile droplet composed of PEG and dextran. With the ability to form micro-sized compartments, our model system has been demonstrated as a powerful reactor to enrich nucleic acids, support flow of genetic information as well as enhance ribozyme cleavage reaction. Although PEG/dextran polymers unlikely existed on the early Earth, we use them as a model system to illustrate evaporation triggered LLPS of neutral polymers. Without loss of generality, our model system of PEG/dextran presents a wide range of neutral macromolecules that tend to segregate in aqueous solutions, which could ubiquitously exist under prebiotic conditions. According to Flory–Huggins theory, in an aqueous solution containing two different neutral polymer species, the Flory interaction parameter between two polymer species is always positive when van der Waals interactions are dominant, implying that an aqueous mixture tends to segregate into different phases to lower the system's free energy[31]. As a result, phase separation commonly exists in many macromolecularly crowded aqueous solutions. Under a simple setting of evaporation, our model system has demonstrated the formation of micro-size compartments via segregative LLPS. Such a mechanism of prebiotic compartmentalization provide insight into how ribozyme-involved reactions might have occurred in the RNA world.

The evaporation process is controlled by the diffusion of water vapor into the ambient environment which, being highly non-uniform, triggers local heterogeneous structures. As we have shown through the phase diagram, the detailed kinetics of this segregative LLPS and the type of emerging structures strongly depends on the exact relative initial composition of the droplet. Further theoretical models based on the interactions of Navier–Stokes equations and Cahn–Hilliard equations are needed to accurately capture time- and composition-dependent phase separation dynamics inside the evaporating droplet. By formulating kinetic pathways on the phase diagram, the increase of TTL with the evaporation is identified, which can be further harnessed to enrich biopolymers inside phase-separated compartments and enhance biochemical reactions. We develop a theoretical framework based on a reaction-diffusion model to account for the enhancement of ribozyme cleavage reaction inside the evaporating ATPS droplet. The main idea is that by introducing RNA partitioning effects and setting boundary conditions for length-dependent RNA compartmentalization, the spatial localization and enrichment of the reactant RNA in dextran-rich compartments induced by segregative LLPS can be well captured. The analysis shows that the dextran-rich compartments have significant advantages over water droplets in both accelerating and promoting ribozyme cleavage, in good agreement with our experimental results. Our work suggests that length-dependent RNA compartmentalization inside phase-separated compartments could have been harnessed by a variety of prebiotic processes.

Compartmentalization via nonassociative phase separation of PEG and dextran has been previously reported as a reactor to enhance ribozyme catalysis, where a reaction rate increase of nearly 70-fold was identified[27]. A recent study demonstrated that RNA self-replication is enhanced fourfold through similar compartmentalization in the PEG/dextran system[73]. Meanwhile, our model system is easy to scale up and highly compatible with wet-dry cycles likely commonplace during early evolution.

An alternative pathway for prebiotic compartmentalization could be achieved inside the complex coacervates that formed through associative phase separation of oppositely charged species. RNA partitioning into the coacervates would be mainly achieved by ion-pairing interactions. However, the highly negatively charged nature of RNA indicates that there would always be complex RNA-polycation interactions that may be unproductive for RNA folding inside the coacervates. Previous studies have identified and tested a variety of complex coacervates formed by polyelectrolytes with different size and sequences. The impact of these coacervates on the prebiotically relevant processes, including RNA partitioning, RNA structural stabilities, and enzymatic reactions, varies and largely relies on the physicochemical properties of charged molecules that trigger phase separation. For example, the partitioning coefficient of poly U15 inside the poly U-spermine coacervate is about 60-fold less than that of poly A15 RNA[74]. Compared with the reaction in the buffer with no coacervate formed, a rate reduction from 13-fold to 60-fold of HHR inside polylysine/carboxymethyldextran coacervates was found[26]. In contrast, a five- to ten-time higher product of HHR cleavage was identified inside the poly-diallyldimethylammonium chloride (PDAC)/poly A11 RNA (rA11) coacervates[25]. Hence, more work on the generalization of the impact of complex coacervates on RNA folding and reaction rates is needed to fully understand how coacervates formed through associative phase separation could serve as primitive compartments. There may also exist a synergistical mechanism, where both associative and nonassociative phase separation could play roles in prebiotic compartmentalization. For example, by the addition of PEG, the RNA oligomer partitioning into the polyU-spermine coacervates was increased by more than two orders of magnitude[75]. Likewise, a more than fivefold enhancement of transcription rate was achieved inside PEG-containing coacervates of cell lysate protein[76]. It is worthwhile to explore new model systems where multiple mechanisms could work synergistically to enhance prebiotically relevant reactions.

There are also implications of our model system for modern-day cells. LLPS has been recognized as an essential cellular compartmentalization strategy in both prokaryotes and eukaryotic cells. Non-membrane-bound liquid droplets rich in protein and/or RNA, also called biomolecular condensates or membraneless organelles, are formed. For example, RNA can be compartmentalized in a variety of membraneless organelles including P-bodies, P granules, the nucleolus, and Cajal bodies[14]. These structures reside in the nonequilibrium environment of living cells and regulate chemical reactions far from equilibrium. Hence, our model system may also lend a deeper understanding to mechanisms of compartmentalization in modern-day cell biology.

## Methods

**Chemicals and Solutions.** The sessile droplet consisted of poly(ethylene glycol) (PEG, BioUltra, $M_w = 8000$, Polydispersityindex :1.22; Sigma-Aldrich) and dextran ($M_w = 10000$, Polydispersityindex :1.2 − 1.7; Aladin) aqueous mixture. The two polymers were dissolved in Milli-Q water. To assist fluorescence imaging, 0.01 mg mL$^{-1}$ fluorescein isothiocyanate-dextran (FITC-dextran, 4 kg mol$^{-1}$, $\lambda_{ex} = 493$ nm, $\lambda_{em} = 517$ nm; Sigma-Aldrich) and 1 mg mL$^{-1}$ Rhodamine B (>95% (HPLC), $\lambda_{ex} = 553$ nm, $\lambda_{em} = 627$ nm; Sigma-Aldrich) were added to the mixture. Nucleic acid sequences of Cy5-DNA, Broccoli DNA template, Hammerhead (and X-Motif) ribozyme DNA template, Hammerhead (and X-Motif) substrate as well as primers used are ordered from Integrated DNA Technology (IDT) and listed in Supplementary Table 2.

**Experimental Setup.** A 0.5 μL droplet was pipetted on a transparent microscope glass surface under room conditions (Temperature = 22 °C, Humidity = 55–65%). The contact angle of the droplet varied between 45° and 25° during the whole evaporation process, measured by bright-field imaging (Nikon) from the side view. The microscope glass slides (ISOLAB GmbH) were used as solid substrates for the sessile droplets. The glass slides were firstly wiped by ethanol wetted tissue to mechanically get rid of contaminants on the surfaces. Then the slides were sonicated in ethanol for 15 min, followed by successively washing with isopropyl alcohol and Milli-Q water, to remove organic contaminants on the surfaces. Then we dried the slides by nitrogen flow and put them into oven (65 °C) for 1 h.

**Microscope observations.** The bright-field images were captured by a high-speed camera (Phantom V9.1, FASTCAM SA4, Photron) coupled with an inverted microscope (Motic AE2000). The fluorescence images were captured by a Nikon Ti2-E Widefield microscopy. For imaging channels of FITC-dextran and Rhodamine B, the excitation wavelength were 475 and 575 nm, respectively. The captured images are processed using ImageJ (NIH) software. Droplet deposits after fully drying on the silica wafer were sputtered with gold before imaging under scanning electron microscope (Hitachi S3400N VP, 20.0 kV).

**BrocT dsDNA generation.** A PCR mixture was assembled containing 1x PCR buffer, 300 μM dNTPs, 400 nM forward and reverse primers, 1 nM template DNA, and 3 units of PFU polymerase (M7745, Promega) in a 100 μL reaction. Thermal cycling consisted of one cycle of 96 °C for 1 followed by 10 cycles of 96 °C for 15 s, 50 °C for 30 s, and 72 °C for 45 s, followed by one cycle of 72 °C for 1 min and hold at 4 °C. The PCR product was purified using a QIAquick PCR Purification Kit (28104, QIAGEN). To generate Cy-5 BrocT dsDNA template, Cy5 labeled forward primer was used in the PCR reaction.

**BrocT RNA aptamer preparation.** IVT mixture (AS3107, Lucigen) was assembled containing 1× transcription buffer, NTPs, DTT, T7 RNA polymerase mix, 50 nM Broccoli dsDNA template, and 200 μM DFHBI-1T. The transcription mix was incubated at 37 °C for 1 h to allow transcription of Broccoli RNA. The ATPS solution is made up of 9 wt% PEG and 4 wt% dextran. The transcription product containing Broccoli RNA aptamers was then added to the ATPS solution with the volume ratio of 1:10. A 0.5 μL droplet of the mixture was pipetted on the glass substrate for evaporation and imaging.

**In vitro transcription.** IVT mixture (AS3107, Lucigen) was assembled containing 1× transcription buffer, NTPs, DTT, T7 RNA polymerase mix, and 200 μM DFHBI-1T. The ATPS solution is made up of 11 wt% PEG and 2 wt% dextran. Then the transcription mixture and 50 nM dsBroccoli DNA template solution were added into the ATPS solution with the volume ratio of 1:1:10. A 5 μL droplet of the mixture was pipetted on the glass substrate for evaporation and imaging.

**Ribozyme cleavage reaction.** Both HHR and X-motif ribozyme are used for the cleavage reaction inside the evaporating droplet. The ribozyme cleavage experiments were carried out with RNAse free reagents in an RNAse free environment. The ATPS solution is made up of 11 wt% PEG and 2 wt% dextran. IVT mixture (AS3107, Lucigen) was assembled containing 1× transcription buffer, NTPs, DTT, T7 RNA polymerase, and 50 nM ribozyme DNA template. The transcription mix was incubated at 37 °C for 1 h to allow transcription of ribozyme RNA. The final mixture is prepared by adding 1 μL of ribozyme substrate (10 μM) and 1 μL of ribozyme RNA transcription product into 38 μL ATPS solution. The ribozyme reaction is started by pipetting a 5 μL droplet of the final mixture on the glass substrate.

**Ribozyme cleavage over time.** The HHR was used to further demonstrate ATPS induced reaction increase under wet-dry cycling conditions. The experiments were carried out with RNAse free reagents in an RNAse free environment. The ribozyme reactions were done with both ATPS and no ATPS buffer systems. The ATPS solution was made up of 11 wt% PEG and 2 wt% dextran in 5 mM Tris pH 7.4, 1 mM MgCl$_2$. The non-ATPS solution was 5 mM Tris pH 7.4, 1 mM MgCl$_2$. The reaction mixture was made up of 38 uL ATPS/non-ATPS solution, 1 μL of ribozyme substrate (10 μM) and 1 μL of ribozyme RNA transcription product. For ATPS and non-ATPS condition, twelve 2-μL aliquots of reaction mixture were made, six aliquots for with wet-dry cycling time points and six aliquots for without wet-dry cycling time-points. The wet-dry cycling samples were incubated in open cap centrifuge tubes in a desiccant filled box to aid evaporation. For the first time-point of 0 h 0.4 μL of RNA PAGE loading dye with 2 mM EDTA was added to the sample and then transferred to −20 °C for storage. Every subsequent time-point (1, 2, 3, 4, and 5 h) 0.4 μL of RNA PAGE loading dye was added to the corresponding time point samples before storage at −20 °C. Additionally 1 μL of water was added to each of the remaining wet-dry cycle samples. After the time course experiments, all samples were loaded onto a 10% TBE PAGE gel for separation before fluorescent imaging analysis at both ex490/em525 and ex556/em573 using a ChemiDoc MP Imaging System (Bio-Rad).

## Data availability

The experimental and numerical data in support of the findings in this study are available within the article and its Supplementary Information, and also from the corresponding author upon request. Source data are provided with this paper.

## Code availability

All numerical codes used to computationally study the reaction-diffusion kinetic models are available from the corresponding author upon request.

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

## Acknowledgements

We thank Dr. Youchuang Chao, Dr. Zhou Liu, Dr. Yang Xiao, Ms. Danyang Ji, and Mr. Fuyun Tan for very helpful discussions. We gratefully acknowledge the computation

assistance in solving the kinetic-diffusion models by Ms. Shu Ma at the Department of Applied Mathematics, The Hong Kong Polytechnic University. This research is supported by the General Research Fund (Nos. 17304017, 17305518, 17127515, and 17163416) and Research Impact Fund (R7072-18), the NSFC Excellent Young Scientists Fund (Hong Kong and Macau) (21922816), the Seed Funding for Strategic Interdisciplinary Research Scheme 2017/18 from the University of Hong Kong, as well as the Sichuan Science and Technology Program (2018JZ0026). H.C.S. is also partially supported by the Croucher Foundation through the Croucher Senior Research Fellowship.

## Author contributions

W.G. and H.C.S. designed the research. W.G., A.K., and Y.Z. performed experiments. W.G. and A.K., analysed the data. W.G., A.K., J.A.T., and H.C.S. had discussions on the results and analysis. W.G., A.K., J.A.T., and H.C.S. wrote the manuscript with comments from all the co-authors.

## Competing interests

The authors declare no competing interests.
