## [Peer Review File · Nature Communications]

REVIEWER COMMENTS

Reviewer #1 (Remarks to the Author):

In this interesting manuscript, the authors demonstrate that the evaporation of a sessile droplet and its concomitant changes in solute concentrations can drive a liquid-liquid phase separation. In addition, Marangoni flows drive small droplets of one of the phases into the center of the sessile droplet, where they coalesce.

Notably, unlike many other recently studied "associative" LLPS systems, the LLPS is based on neutral components (PEG and dextran). This appears to have the advantage that genetic molecules (DNA/RNA) can be localized in phase separated droplets, but still retain their functionality. By contrast, associative LLPS (based on charged polymers) have been shown to rather inhibit reactions involving DNA and RNA.

The authors argue that their system may be relevant both to the understanding of compartmentalization in cells as well as for the emergence of primordial cells.

The paper is very well written and the results are nicely presented. This reviewer mainly has reservations in terms of the "relevance" for primordial or extant biology, which probably can be addressed by cautiously rephrasing the manuscript in part.

Specific points:

- The authors claim "This simple yet robust system can capture both the macromolecularly crowded interior environment and the non-equilibrium features of extant cells. Hence, the model may be important for compartmentalization in early molecular evolution."

It is not completely clear how the model can capture *both* features of extant cells and early molecular evolution. It seems that – throughout the manuscript – some points more relevant to early evolution (wet-dry cycles, concentration of nucleic acids, etc.) are mixed with other features that are more relevant maybe to modern biochemistry (use of RNA polymerase).

The authors also claim "These similarities make it possible to bridge prebiotic LLPS to protocells with subcellular membraneless compartments by using our model system." One could argue that either just similar (physicochemical) processes play a role in both cases – or, that there is a continuous evolutionary thread from one to the other. The latter seems to contradict that apparently cellular life started with prokaryotes with different types of "compartmentalization" rather than LLPS?

Also - how relevant is the PEG/dextran system? What could potentially take its place either in prebiotic conditions or in extant cells?

- Regarding the Marangoni flow towards the center – is there a counterflow of fluid in the center of the sessile droplet towards the rim?

- "we conclude that transcription exclusively occurs in the phase-separated compartments". It is not completely clear whether this is true – maybe transcription occurs everywhere – or at the interface of the droplets (depending on whether the polymerase can enter the droplets) – and the RNA product is just quickly incorporated into the droplets? This might be answered by labeling the DNA template and maybe also the polymerase and check their localization.

- again the formulation "reactor that simulates the flow of genetic information under prebiotic conditions" is slightly problematic as you use modern components (DNA, RNA polymerase, ...), which may not be prebiotic.

- in the HHR experiments it is not completely obvious whether the activity is really enhanced – the signal also rises in the "no ATPS" group, but it is distributed in the whole droplet and not localized to the phase-separated dextran-rich droplet. Maybe the total amount of product (integrated over the droplet) is the same? What would be the origin of such an enhancement – larger local concentration of HHR and substrate? One could argue that enhanced concentration also leads to more product inhibition?

- Could you perform wet-dry cycles experimentally with this system and couple them, e.g., to some kind of replication process?

Reviewer #2 (Remarks to the Author):

The manuscript from Guo et al. describes the formation of membraneless compartments from non-associative phase separation. The work includes an evaporative flux model as well as a credible model for local flows driven by Marangoni forces. Additionally, the manuscript describes the localization of DNA and RNA, highlighting the role of phase separation in in vitro transcription as well as ribozyme activity.

The manuscript is well written, and the figures clearly convey the key messages, focusing on the potential role of non-associative phase separation in early evolutionary development. The experiments were thoughtfully constructed, and the results clearly show that the compartmentalization of DNA and RNA significantly influences kinetics of biological processes. The key weakness in the manuscript is lack of connection between the proposed models presented in the first part of the manuscript and the rates of the processes presented in the second part of the narrative. That said, the work will likely have a significant impact in a multidisciplinary community of scientists, and this reviewer believes that the work will be well received by the readers of Nature Communication. Some comments below are included that may improve the manuscript.

Comments:

- The stated aim of this work is to use the evaporating droplet as a 'model' for prebiotic compartmentalization, and the models presented cleverly focus on the 'pathway' for phase separation and flow within the droplet. That said, these transport models should give one the ability (even in heterogeneous systems) to examine local concentrations. Coupling the evaporative models with a set of simple heterogeneous 'reaction' kinetic models for the DNA and RNA processes would significantly strengthen the manuscript.

- Outside of invoking the Marangoni force to understand the flow, the role of the interfaces in this system are largely neglected. From the movies and the figures, it appears as if the formation of the compartments initially occurs at the solid-liquid interface, liquid-air interface or three phase contact line. For example, many of the images highlight the 'nucleation' of the smaller compartments at the three-phase contact line. Additionally, some of the dextran-rich compartments appear to 'pin' to the solid liquid-interface and ripen before moving inward.

- The manuscript states that the "small nucleated droplets ranges from hundreds of nanometers to several microns", which is clear from the optical microscopy. Are there nucleating droplets observed that are smaller? The authors include some SEM images of dehydrated compartments in the Supporting online material, but it is unclear if these images were done at a time that would include the nascent compartments.

- The discussion section is more of a summary. For greater impact, the authors could use this section to highlight how these findings deepen the understanding in the areas of coacervation and origins of life. A quantitative comparison could be made between models used by others as well as rates observed in similar systems.

- The figures clearly present ideas and results, but the captions could be more descriptive. For example, in figure one: (a) the caption could indicate the purpose of the 'rock' and (b1 and b2) the caption could indicate the meaning of the red star.

- Are the PEG and dextran molecules monodispersed? Can polydispersity be included in the Methods section?

Reviewer #3 (Remarks to the Author):

This manuscript delves into the phase separation mechanisms of an evaporating aqueous droplet containing small amounts of Polyethylene glycol and dextran. Then the authors demonstrate the nucleic acids partitioning, active transcription and enhanced ribozyme cleavage inside the newly formed Dextran phase and develop the hypothesis based on their observations that aqueous two-phase separation might have played a role in early molecular evolution. The strength of this paper lies in its experimental simplicity, compelling data and diligent explanation of the mechanism of the pattern-directed phase separation in their binary polymer mixture droplets. The unique scope of this manuscript that brings together the subject of aqueous two-phase separation (ATPS) mechanisms and their role as plausible primordial reactors could spike the interest of diverse audience of Nat. Comm. ranging from molecular biologists, microbiologists and biochemists to evolutionary biologists. The experimental concept is novel, and the findings certainly have merit. However, there are certain caveats in the manuscript, mainly in the writing that needs addressing before it can be considered for publication.

Some of my major concerns/comments are listed below:

1. The manuscript fails to provide critical statistical information such as sample size, statistical significance to the differences in measurements acquired etc. It also falls short on indicating the future direction of the study or the impact of these findings on ongoing ATPS researches.
2. There appears to be a disjoint in the flow of the manuscript. A considerable length of the paper talks in detail about the kinetic pathways of concentration dependent phase separation of polymers but it is not clear how the first half of the paper plays a role in determining the phase compositions used in all the nucleic acid studies.
 - a. Did the authors see any difference in the phenomenon like partitioning, transcription or cleavage in regime I vs regime II?
 - b. What was the ATPS phase concentrations in each of these studies?
3. The paper needs more discussion on why, despite having very similar evaporation rates, the phase separation might enhance ribozyme cleavage or DNA transcription compared to evaporating water droplet? Is the increased relative concentration of substrates in Dextran rich compounds comparable or significantly higher than that in single phase evaporation?
4. The current discussion section reads more like a conclusion and therefore should be designated as such. Moreover, the paper could benefit from a separate discussion focusing on different ways the phase separation might impact DNA and RNA activities, compare their findings to literature, discuss the impact of their study on contemporary studies and provide implications of these results on further studies.

Some additional minor comments are as follows:

1. It would be easy for the readers to see the fitness of theoretical models if figures S2 and S4 also plotted theoretical and (some) measured values respectively.
2. The authors should be attentive not to miss defining all the variables use in the equations and their derivations. (E.g. some variables like τ , RH were missed)
3. What do the pink squares in fig 2 (a) represent?
4. In supplementary text S2, figure S4 is mis-indicated as figure S5.
5. At that time point was the droplet size and count measured in fig S6 (b)? Weren't the drops coalescing and thus increasing in size with time?

REVIEWER COMMENTS

Reviewer #1 (Remarks to the Author):

In this interesting manuscript, the authors demonstrate that the evaporation of a sessile droplet and its concomitant changes in solute concentrations can drive a liquid-liquid phase separation. In addition, Marangoni flows drive small droplets of one of the phases into the center of the sessile droplet, where they coalesce.

Notably, unlike many other recently studied “associative” LLPS systems, the LLPS is based on neutral components (PEG and dextran). This appears to have the advantage that genetic molecules (DNA/RNA) can be localized in phase separated droplets, but still retain their functionality. By contrast, associative LLPS (based on charged polymers) have been shown to rather inhibit reactions involving DNA and RNA.

The authors argue that their system may be relevant both to the understanding of compartmentalization in cells as well as for the emergence of primordial cells. The paper is very well written and the results are nicely presented. This reviewer mainly has reservations in terms of the “relevance” for primordial or extant biology, which probably can be addressed by cautiously rephrasing the manuscript in part.

We thank the reviewer for the helpful comments. All comments have been addressed through further experimentation and rewriting to improve the manuscript as we specify below.

Specific points:

1. The authors claim “This simple yet robust system can capture both the macromolecularly crowded interior environment and the non-equilibrium features of extant cells. Hence, the model may be important for compartmentalization in early molecular evolution.”

It is not completely clear how the model can capture *both* features of extant cells and early molecular evolution. It seems that – throughout the manuscript - some points more relevant to early evolution (wet-dry cycles, concentration of nucleic acids, etc.) are mixed with other features that are more relevant maybe to modern biochemistry (use of RNA polymerase).

We thank the reviewer for allowing us to clarify on the relevance of our work on the early molecular evolution and intracellular activities. Firstly, the evaporating droplets can trigger segregative liquid-liquid phase separation of neutrally charged polymers, forming immiscible membraneless compartments that can encapsulate nucleic acids. This compartmentalization is considered to be of vital importance to the emergence of life under the “RNA world” hypothesis. Secondly, extant cells are densely packed with macromolecules. For example, a total macromolecule concentration of over 300 mg/mL in *E. coli* is achieved¹. This macromolecular crowding affects the physicochemistry of the cytoplasm, which, in turn, affects intracellular activities such as biomolecule mobility, protein folding and polyelectrolyte association². Due to the high viscosity of the densely packed macromolecules, it is difficult to create protocells with such a high polymer concentration through conventional formulations. Living cells are highly non-equilibrium systems³, in which a majority of dynamic processes essential for life, such as cytoskeleton reorganization⁴ and messenger RNA localization,⁵ occur. In our work, with the evaporation setting, the non-equilibrium cell-mimicking environments with macromolecular crowding can be easily achieved inside our model system. To summarize, while its role for prebiotic compartmentalization is emphasized, our model also provides a platform for mimicking intracellular functions. The LLPS demonstrated within our droplet compartments is similar to those observed in membraneless organelles inside living cells, and

thus is promising in connecting the origin of life community with those on membraneless protocells. The sentence mentioned in this comment (at the end of the **Introduction** part in the main text) has been revised as follows:

“Our model can validate non-associative phase separation in moderate conditions with no heating, cooling or osmotic shock needed. Thus, we elucidate a plausible pathway for prebiotic compartmentalization via non-associative phase separation. Furthermore, the simple yet robust system can capture both the macromolecularly crowded interior environment and the non-equilibrium features of extant cells, providing cell-mimicking conditions essential for a variety of intracellular functions.”

We also added the relevant descriptions in the **Discussion** part of the main text:

“The RNA world hypothesis is a prevailing model for origin of life research, where RNA acted both as a catalyst and as a genetic storage molecule. In the RNA world, prebiotic compartmentalization would have been essential to enrich nucleic acids from the dilute primordial environment to a sufficient concentration for reactions and molecular evolution. Here, we have presented a model system for prebiotic compartmentalization strategies by harnessing segregative LLPS within an evaporating sessile droplet composed of PEG and dextran. With the ability to form micro-sized compartments, our model system has been demonstrated as a powerful reactor to enrich nucleic acids, support flow of genetic information as well as enhance ribozyme cleavage reaction.” and

“There are also implications of our model system for modern-day cells. LLPS has been recognized as an essential cellular compartmentalization strategy in both prokaryotes and eukaryotic cells. Non-membrane-bound liquid droplets rich in protein and/or RNA, also called biomolecular condensates or membraneless organelles are formed. For example, RNA can be compartmentalized in a variety of membraneless organelles including P-bodies, P granules, the nucleolus and Cajal bodies. These structures reside in the non-equilibrium environment of living cells and regulate chemical reactions far from equilibrium. Hence, our model system may also lend a deeper understanding to mechanisms of compartmentalization in modern-day cell biology.”

2. The authors also claim “These similarities make it possible to bridge prebiotic LLPS to protocells with subcellular membraneless compartments by using our model system.” One could argue that either just similar (physicochemical) processes play a role in both cases – or, that there is a continuous evolutionary thread from one to the other. The latter seems to contradict that apparently cellular life started with prokaryotes with different types of “compartmentalization” rather than LLPS?

Besides the potential role in prebiotic compartmentalization through segregative LLPS, our model system also captures the non-equilibrium and macromolecular crowding features of the nucleus and cytoplasm. The compartmentalization and LLPS together enables significantly enhanced kinetics of a variety of physicochemical processes similar to those essential for living cells. This is consistent with recent reports where LLPS is harnessed by prokaryotes to generate subcellular compartments (PNAS, 2020 117 (31) 18540-18549)⁶ This has been introduced in the modified comments mentioned in response to previous comment.

Also -How relevant is the PEG/dextran system? What could potentially take its place either in prebiotic conditions or in extant cells?

This is indeed an important comment that we hope to clarify. We do not suggest that PEG and dextran polymers were present on the early Earth; however, we use them as a model system to illustrate evaporation triggered LLPS of neutral polymers. Phase separation commonly exist in a variety of macromolecularly crowded polymer solutions and provides a plausible route to compartmentalization. The findings we presented are not confined to the model system of PEG/dextran, and can be extended to a wide range of neutral polymers that tend to segregate in aqueous solutions, which may exist under prebiotic conditions. Based on Flory-Huggins theory⁷, in an aqueous solution containing two different neutral polymer species, *A* and *B*, with the structures of -A-A-A-...-A-A- and -B-B-B-...-B-B- respectively, the Flory interaction parameter between *A* and *B*, which can be denoted as χ_{AB} , is always positive ($\chi_{AB} > 0$) when van der Waals interactions are dominant. A positive χ_{AB} implies that *A* and *B* tend to segregate into two phases (*A*-rich phase and *B*-rich phase, respectively) to lower the system free energy. Hence, strong segregation effects are not specifically confined in PEG/dextran systems, but also widely exist in many other water-soluble polymer systems and polymer/salt systems, as reported in previous studies (J. Am. Chem. Soc. 2012, 134, 22, 9094–9097)⁸ and reviewed in a recent review paper (Chem. Soc. Rev., 2020, **49**, 114-142)⁹. PEG and dextran are chosen for their extensive use as crowding reagents in many biochemical assays as well as protocells (Curr. Opin. Struct. Biol. 20, no. 2 (2010): 196-206)¹⁰. This has been described in the **Discussion** part of the main text:

“Although PEG/dextran polymers unlikely existed on the early Earth, we use them as a model system to illustrate evaporation triggered LLPS of neutral polymers. Without loss of generality, our model system of PEG/dextran presents a wide range of neutral macromolecules that tend to segregate in aqueous solutions, which could ubiquitously exist under prebiotic conditions. According to Flory-Huggins theory, in an aqueous solution containing two different neutral polymer species, the Flory interaction parameter between two polymer species is always positive when van der Waals interactions are dominant, implying that an aqueous mixture tends to segregate into different phases to lower the system free energy⁷. As a result, phase separation commonly exists in many macromolecularly crowded aqueous solutions.”

3. Regarding the Marangoni flow towards the center – is there a counterflow of fluid in the center of the sessile droplet towards the rim?

The reviewer is correct in pointing out that a counterflow of fluid is present to flow from the center of the sessile droplet towards the rim, to compensate for the water loss at the rim due to the high evaporation rate here. For a sessile droplet with no or negligible surfactant components, such as a pure water droplet, the capillary flow from center to the rim is dominant over the evaporation process. With the addition of water-soluble surfactants, Marangoni stress will appear with the concentration gradient of the surfactant and alter the fluid dynamics inside the droplet. In our experiments, the capillary flow is dominant and obvious at the initial stage of droplet spreading after the droplet is pipetted onto the glass substrate. However, as we discussed in the supporting information (**S3. Marangoni stress calculation**), the Marangoni stress grow quickly with the evaporation, generating Marangoni flow that moves from the droplet rim to the center and thus suppresses the capillary flow, as shown in **Figure 1 (Video S1 and S2 as well)**. As we have described in the main text:

“From pattern evolutions of LLPS in both regimes, we observe that the phase-separated dextran-rich compartments always move towards the center of the sessile droplet, rather than being advected to droplet edge by outwards capillary flows, as in systems exhibiting the so-called “coffee-ring effects”.” (Page 10 of 28) and

“...the Marangoni flow driven by the surface tension differences plays a dominant role in the movement of phase-separated compartments inside the sessile droplet.” (Page 11 of 28).

4. “we conclude that transcription exclusively occurs in the phase-separated compartments”. It is not completely clear whether this is true – maybe transcription occurs everywhere – or at the interface of the droplets (depending on whether the polymerase can enter the droplets) – and the RNA product is just quickly incorporated into the droplets? This might be answered by labeling the DNA template and maybe also the polymerase and check their localization.

We thank the reviewer for this critical comment and the helpful suggestion. We have added one more experiment by using Cy-5 labelled BroccoliT DNA template and observing the *in vitro* transcription inside the evaporating ATPS droplet, as shown in **Fig. S16** in the supporting information. The results show that DNA will localize into dextran-rich compartments during the evaporation. The fluorescence signals from RNA-fluorophore complexes can only be observed inside the dextran-rich compartments, confirming our claim that transcription reaction exclusively occurs in the phase-separated dextran-rich compartments. We also added the corresponding descriptions in the main text, which now reads:

“To confirm that the transcription does occur inside the phase-separated dextran-rich droplets, we added Cy-5 labelled BroccoliT DNA template into the ATPS droplets and observed the localization of both DNA template and transcribed RNA aptamers. The results show that DNA localizes into dextran-rich droplets during evaporation and the fluorescence of RNA-fluorophore complexes can be only observed inside the dextran-rich compartments (SI Appendix, Fig. S16).” (Page 14 of 28)

5. Again the formulation “reactor that simulates the flow of genetic information under prebiotic conditions” is slightly problematic as you use modern components (DNA, RNA polymerase, ...), which may not be prebiotic.

We thank the reviewer for this suggestion that makes our statement more precise. We agree that compared with ribozyme, DNA and RNA polymerase may not be prebiotic. Our model system is proposed as a cell-mimicking microenvironment with macromolecular crowding features, and thus has potentials in achieving the prebiotic compartmentalization for the relevant biochemical reactions to occur under cellular conditions. As a demonstration, we performed DNA transcription experiments inside the evaporating droplets, which suggests that the phase-separated compartments can act as reactors that allow the flow of genetic information. The demonstration suggests that other molecules relevant to the RNA world hypothesis, such as RNA and proteins observed in early earth, can also be applied based on the compartmentalization and cellular mimicking shown in our model system (part of the reply to comment 1 and 2). We have also revised the sentence in the main texts accordingly, which now reads:

“Hence, our model system can provide a cell-mimicking environment that supports the flow of genetic information. The compartmentalization of functional biopolymers provides a plausible strategy to assemble molecules relevant to the RNA world hypothesis under prebiotic compartmentalization conditions, and could have played a role in the evolution of informational polymers on the early Earth.” (Page 14 to 15 of 28)

6. In the HHR experiments it is not completely obvious whether the activity is really enhanced – the signal also rises in the “no ATPS” group, but it is distributed in the whole droplet and not localized to the phase-separated dextran-rich droplet. Maybe the total amount of product (integrated over the droplet) is the same? What would be the origin of such an enhancement – larger local concentration of HHR and substrate? One could argue that enhanced concentration also leads to more product inhibition?

We thank the reviewer for this important question. To quantify the activity, we also measured the integrated fluorescence intensity over the whole droplet as a function of time, and our results suggest that ribozyme activity in the dextran-rich compartments is enhanced more than 3-fold compared with that in the water droplet, as shown in **Fig. S12**. Additionally, our PAGE gel analysis of entire ribozyme cleavage reactions over time indicates an ATPS-induced enhancement, as shown in **Fig. S14**. As the entire reaction is analyzed, it is a net effect of the reaction and not just a localised effect.

Both HHR and substrate are partitioned into dextran-rich phase and thus have a higher local concentration inside the compartments, which is favorable for the acceleration of the reaction rate. However, the origin of the enhanced reaction rates is not only due to the increase of the local concentrations of HHR and substrates, but also because of the selective partitioning strength of RNA with different lengths. As reported in the previous study (*Nat. Commun.* **5**, 4670 (2014))¹¹, the partitioning coefficient of HHR (43-nt length) in ATPS is about 0.006 ($[RNA]_{PEG-rich}/[RNA]_{Dextran-rich} = 0.06$), which means HHR concentration in dextran-rich phase is more than 100 times higher than that in PEG-rich phase. For the substrate (14-nt length), the partitioning coefficient is about 0.57, hence, the substrate concentration in the dextran-rich phase is about 2 times higher than that in the PEG-rich phase. For the two products (6-nt & 8-nt length, respectively), the partitioning coefficients are slightly higher than 0.7; consequently, the products only have a slightly higher concentration in dextran-rich phase than in PEG-rich phase. This length-dependant selective RNA partitioning strength in ATPS ensures the dextran-rich compartments can serve as a permeable reactor for RNA products of short chains to leave the droplet compartments, while maintaining the high enrichment of HHR inside. This function further enhances the reaction rate and drives the reaction towards completion.

To gain insights on the mechanism of the enhanced ribozyme cleavage by segregative phase separation, we have derived a reaction-diffusion model to explain the enhancement of reaction rate inside the phase-separated dextran-rich droplets (see **equation (3) and (4)** in the main text, as well as **SI Appendix, S4**). We consider both “no ATPS” group and “ATPS” group, by introducing the RNA partitioning effects inside dextran-rich phase and setting up different boundary conditions. The results show that, with the ability to form spatial localization and enrichment of the reactants, the dextran-rich compartments formed through segregative phase separation have significant advantages over water droplets in both accelerating and promoting ribozyme cleavage (see **Figure 5 (e)** in the main text and **Fig. S17** in the SI Appendix), consistent with our experimental observations.

7. Could you perform wet-dry cycles experimentally with this system and couple them, e.g., to some kind of replication process?

The reviewer has raised a significant and important point. Indeed, this work was inspired in part by the wet-dry cycles, which have been demonstrated to contribute to the production of nucleosides and polynucleotides on the early Earth. To further emphasize this, we have added a group of experiments by coupling ribozyme cleavage in phase separating ATPS and wet-dry cycles, as shown in **Figure S14**. The results show that the reaction rate was increased over 3 times for the ATPS and wet-dry cycling condition when compared to the without ATPS, without wet-dry cycling condition. In addition, a recent study has investigated the effects of wet-dry cycles on associative phase separation (*Nat. Commun.* **11**, 5423 (2020))¹². The results suggest that the phase behaviour and functions of complex coacervates can be significantly altered by wet-dry cycling. Taken together, our model system has demonstrated a prebiotically plausible pathway for segregative phase separation, which is highly compatible with wet-dry cycling processes.

Reviewer #2 (Remarks to the Author):

The manuscript from Guo et al. describes the formation of membraneless compartments from non-associative phase separation. The work includes an evaporative flux model as well as a credible model for local flows driven by Marangoni forces. Additionally, the manuscript describes the localization of DNA and RNA, highlighting the role of phase separation in *in vitro* transcription as well as ribozyme activity.

The manuscript is well written, and the figures clearly convey the key messages, focusing on the potential role of non-associative phase separation in early evolutionary development. The experiments were thoughtfully constructed, and the results clearly show that the compartmentalization of DNA and RNA significantly influences kinetics of biological processes. The key weakness in the manuscript is lack of connection between the proposed models presented in the first part of the manuscript and the rates of the processes presented in the second part of the narrative. That said, the work will likely have a significant impact in a multidisciplinary community of scientists, and this reviewer believes that the work will be well received by the readers of Nature Communication. Some comments below are included that may improve the manuscript.

We thank the reviewer for reviewing our work. We appreciate the reviewer's positive evaluation. We have addressed all comments through further experimentation and redrafting.

Comments:

1. The stated aim of this work is to use the evaporating droplet as a 'model' for prebiotic compartmentalization, and the models presented cleverly focus on the 'pathway' for phase separation and flow within the droplet. That said, these transport models should give one the ability (even in heterogeneous systems) to examine local concentrations. Coupling the evaporative models with a set of simple heterogeneous 'reaction' kinetic models for the DNA and RNA processes would significantly strengthen the manuscript.

We thank the reviewer for this insightful comment. As suggested by the reviewer, a reaction-diffusion model for the ribozyme cleavage reaction is added to the manuscript. With the reaction-diffusion model, the kinetics of ribozyme cleavage and the enhancement rate inside phase separated compartments are well described.

As we have introduced in the reply to the comment 6 of the Reviewer 1, briefly, the evaporation triggers the formation of phase separated dextran-rich compartments inside the sessile droplet. DNA and RNA molecules localize into these compartments due to partitioning effects. This localization gives an increase in DNA/RNA concentrations inside the compartments, providing a driving force to accelerate reaction rates. Moreover, in the ribozyme cleavage reaction, the sequence lengths of ribozyme, substrate and products are different. Combining with the length-dependent partitioning strengths of nucleic acids in ATPS, the length difference makes the phase separated compartment a favorable reactor to concentrate long-sequence ribozyme and release short-sequence products, which further inhibit the overaccumulation of products inside the compartments. The length-dependent partitioning strengths can be modelled by introducing RNA partitioning effects of dextran-rich phase and setting different boundary conditions of the reaction-diffusion kinetic model. The details of the model are described in the main text and supporting information (see **equation (3) and (4)** in the main text, as well as **SI Appendix**,

S4). Our model results show that (see **Figure 5 (e)** in the main text and **Fig. S17** in the SI Appendix), owing to the spatial localization and enrichment of the reactants, the dextran-rich compartments formed through segregative phase separation have significant advantages over water droplets in both accelerating and promoting ribozyme cleavage.

2. Outside of invoking the Marangoni force to understand the flow, the role of the interfaces in this system are largely neglected. From the movies and the figures, it appears as if the formation of the compartments initially occurs at the solid-liquid interface, liquid-air interface or three phase contact line. For example, many of the images highlight the ‘nucleation’ of the smaller compartments at the three-phase contact line. Additionally, some of the dextran-rich compartments appear to ‘pin’ to the solid liquid-interface and ripen before moving inward.

This is indeed an important comment. To reveal the role of surfaces and interfaces in phase-separated compartment formation, we captured images at the beginning of the evaporation near the three-phase contact line, as shown in panel (c) and (d) above (Scale bar: 20 μm). Initially, the ATPS droplet has a homogenous single-phase component, as shown in the image sequences above. Shortly after the beginning of the evaporation, phase separation occurs at three-phase contact line of the droplet, owing to the higher evaporation here, resulting in the formation of submicron-sized structures. Following the evaporation process, for the droplet in regime 1, there are lobe-shaped dextran-rich phase formed near the rim, while dispersed micron-sized droplets are developed for the droplet in regime 2, as sketched in panel (a)-(b) above. These phase-separation-induced structures move towards the centre of the sessile droplet by Marangoni flow, with a phase separation front (PSF) that distinguishes the single-phase region and phase-separated region. For both regimes, the nucleation of dextran-rich domains near the three-phase contact line is continuously triggered once the polymer concentration surpasses the binodal curve. These nucleated tiny domains structures quickly coalesce and move towards the droplet centre, leading to further “thickening” of the dextran-rich phase in regime 1 and the

growth of dextran-rich droplets in regime 2. These results show that the higher evaporation rate near the three-phase contact line is the major driving force to trigger phase separation and control the pattern evolution at early times, though the latter is also sensitive to droplet initial compositions. We note that similar dynamics has been previously studied during the nucleation of oil droplets near the three-phase contact line of an evaporating “Ouzo” droplet¹³. We have added the figure and discussions above into supporting information, shown as **Figure. S6**. The main text now reads:

“For both regimes, the nucleation of dextran-rich compartments near the three-phase contact line is continuously triggered once the polymer concentration surpasses the binodal curve (Fig. S6 (b) and (c)). In regime 1, the dextran-rich compartments coalesce quickly after nucleation, forming lobe-shaped domains of dextran-rich phase (Figure 1(c2)). In regime 2, these small compartments remain as dispersed droplets (Figure 1(d2)).” (Page 9 of 28).

The ‘pinning’ of the dextran-rich compartments at the rim can be observed during droplet evaporation process, especially at the late stage of the evaporation, as shown in **Figure 1**, **Figure 3** and **Figure 5** in the main text. We attribute this ‘pinning’ effect to the reduced droplet height that inhibits Marangoni convection. Specifically, as shown in panel (a) above (Scale bar: 100 μm), at the early stage, the sessile droplet (0.5 μL) has a relatively large height of about 200 μm , leaving sufficient space for the development of the inward Marangoni flow. As a result, most of the nucleated droplets near the three-phase contact line can be convected to droplet centre. Further evaporation will cause the decrease of droplet height, until only a thin liquid film is left near the three-phase contact line. At this time point, it is difficult for dextran-rich droplets near the rim to move further. These droplets stop moving inward but only keep growing as they coalesce with the nucleated droplets from phase separation of the liquid film, as shown in panel (b) above (Scale bar: 500 μm), which is also part of **Figure 1** (when $t = 200$ s) in the main text. Finally, just before the end of the evaporation process, the three-phase contact line retracts and leaves these dextran-rich droplets drying with the polymer deposit. The formation of the polymer deposit from drying of the droplet or liquid film also involves rich dynamics and some typical cases have been investigated by Doi and co-workers^{14,15}.

We have added the figure and the above discussions into supporting information, shown as **Figure. S6 (e) - (f)**. Correspondingly, the main text now reads

“The ‘pinning’ of the dextran-rich droplets near the rim is observed before the drying of the sessile droplet (Figure 1 (c) and (d)), which is attributed to the reduced droplet height and thus the confined space for the inhibition of inward Marangoni convection (SI Appendix, Fig. S6 (e) and (f)).” (Page 11 to 12 of 28).

3. The manuscript states that the “small nucleated droplets ranges from hundreds of nanometers to several microns”, which is clear from the optical microscopy. Are there nucleating droplets observed that are smaller? The authors include some SEM images of dehydrated compartments in the Supporting online material, but it is unclear if these images were done at a time that would include the nascent compartments.

In our model system, the diameter of the smallest droplets observed by optical microscopes is about 500 nm, as shown in **Fig. S7 (c)**. Submicron-sized droplets can also be observed near the droplet rim by phase contrast microscopy with a 40x objective, as shown in the following image. Small droplet patterns with submicron size can also be found in the SEM images of droplet depositions after the evaporation process is completed (**Fig. S1 (d)-(f)** and **Fig. S7 (d)**).

As evidenced theoretically and experimentally, the smallest droplet nucleated from polymer phase separation has a size of a few tens of nanometers, known as the droplet “nuclei”. For example, the phase-separated “nuclei” in the ternary mixture of polyethyl-butylene (dPE), polymethylbutylene (hPM), and apolymethylbutylene-block-polyethylbutylene copolymer (hPM-hPE) has a size ranging from 20 nm to 50 nm at the early stage of phase separation, which contains only a few polymer chains (Phys. Rev. Lett. **77**, 3847, (1996))¹⁶. In our model system, if we assume that there are no more than 10 polymer chains inside the phase-separated “nuclei” in PEG/dextran, the size of the nuclei could also be in the range of a few tens of nanometers, given that the polymer gyration radius of PEG and dextran is a few nanometers. To the best of our knowledge, the “nuclei” size in aqueous PEG/dextran solutions has not yet been systematically studied. Nevertheless, our results show that the model system has provided a simple method to obtain submicron-sized all-aqueous droplets, which are difficult to make by other techniques, including microfluidics. These droplets have similar sizes as some intracellular organelles and are promising for constructing artificial protocells and organelles. We have included the following images and discussions into the supporting information (**Fig S7 (e)**). The content in the main text now reads

“Droplet “nuclei” with a smaller size than submicron-sized dextran-rich droplets in Fig. S7 could be expected at the initial stage of polymer phase separation (Ref: 59).” (Page 11 of 28).

4. The discussion section is more of a summary. For greater impact, the authors could use this section to highlight how these findings deepen the understanding in the areas of coacervation and origins of life. A quantitative comparison could be made between models used by others as well as rates observed in similar systems.

We thank the reviewer for this critical comment. In response, we have added a summary in the **Discussion** section, quantitatively comparing phase separation and complex coacervates as well as their roles in prebiotically relevant processes in recent studies, as follows:

“Compartmentalization via non-associative phase separation of PEG and dextran has been previously reported as a reactor to enhance ribozyme catalysis, where a reaction rate increase of nearly 70-fold was identified. A recent study demonstrated that RNA self-replication is enhanced 4-fold through similar compartmentalization in the PEG/dextran system.(Ref 77) Meanwhile, our model system is easy to scale up and highly compatible with wet-dry cycles likely commonplace during early evolution.

An alternative pathway for prebiotic compartmentalization could be achieved inside the complex coacervates that formed through associative phase separation of oppositely charged species. RNA partitioning into the coacervates would be mainly achieved by ion-pairing interactions. However, the highly negatively charged nature of RNA indicates that there would always be complex RNA-polycation interactions that may be unproductive for RNA folding inside the coacervates. Previous studies have identified and tested a variety of complex coacervates formed by polyelectrolytes with different size and sequences. The impact of these coacervates on the prebiotically relevant processes, including RNA partitioning, RNA structural stabilities and enzymatic reactions, varies and largely relies on the physicochemical properties of charged molecules that trigger phase separation. For example, partitioning coefficient of poly U15 inside the poly U-spermine coacervate is about 60-fold less than that of poly A15 RNA. Compared with the reaction in the buffer with no coacervate formed, a rate reduction from 13-fold to 60-fold of hammerhead ribozyme inside polylysine/carboxymethyl-dextran coacervates was found. In contrast, a 5- to 10-time higher product of hammerhead ribozyme cleavage was identified inside the polydiallyldimethylammonium chloride (PDAC) /poly A11 RNA (rA11) coacervates. Hence, more work on the generalization of the impact of complex coacervates on RNA folding and reaction rates is needed to fully understand how coacervates formed through associative phase separation could serve as primitive compartments. There may also exist a synergistical mechanism, where both associative and non-associative phase separation could play roles in prebiotic compartmentalization. For example, by the addition of PEG, the RNA oligomer partitioning into the polyU-spermine coacervates was increased by more than two orders of magnitude. Likewise, a more than 5-fold enhancement of transcription rate was achieved inside PEG-containing coacervates of cell lysate protein. It is worthwhile to explore new model systems where multiple mechanisms could work synergistically to enhance prebiotically relevant reactions.”

5. The figures clearly present ideas and results, but the captions could be more descriptive. For example, in figure one: (a) the caption could indicate the purpose of the ‘rock’ and (b1 and b2) the caption could indicate the meaning of the red star.

As suggested by the reviewer, we have modified all figure captions accordingly to make them more descriptive. Specifically, for the caption highlighted in this comment, the rock serves as the substrate and provides a silica-rich surface that support various prebiotic reactions such as the polymerization of the polynucleotides. The red stars denote the initial compositions of sessile droplets: (b1) shows a composition of 5 wt% PEG and 10 wt% dextran, and (b2) shows 9 wt% PEG and 4 wt% dextran. These descriptions have been added into the caption of **Figure 1** in the main text.

6. Are the PEG and dextran molecules monodispersed? Can polydispersity be included in the Methods section?

The PEG and dextran are used without any further purifications after purchase. According to the commercial information, the polydispersity of the PEG and dextran molecules are 9000~10000 (average: 8000) and 9000~11000 (average: 10000), respectively. We have added this information to the **Methods** section. The evaporation triggered segregative phase separation phenomenon in our model system is robust for PEG and dextran over a wide range of molecular weight, as shown in the Figure R1.

Figure R1. Phase-separated pattern evolution inside an evaporating sessile droplet composed of PEG and dextran with different molecular weight. The droplet has a composition of 3 wt% PEG 4000, 3 wt% PEG 8000, 2 wt% PEG 20000, 1 wt% dextran 10000, and 1 wt% dextran 40000 (so that the final composition is 8 wt% PEG and 2 wt% dextran, respectively). The scale bar is 500 μm.

Reviewer #3 (Remarks to the Author):

This manuscript delves into the phase separation mechanisms of an evaporating aqueous droplet containing small amounts of Polyethylene glycol and dextran. Then the authors demonstrate the nucleic acids partitioning, active transcription and enhanced ribozyme cleavage inside the newly formed Dextran phase and develop the hypothesis based on their observations that aqueous two-phase separation might have played a role in early molecular evolution. The strength of this paper lies in its experimental simplicity, compelling data and diligent explanation of the mechanism of the pattern-directed phase separation in their binary polymer mixture droplets. The unique scope of this manuscript that brings together the subject of aqueous two-phase separation (ATPS) mechanisms and their role as plausible primordial reactors could spike the interest of diverse audience of Nat. Comm. ranging from molecular biologists, microbiologists and biochemists to evolutionary biologists. The experimental concept is novel, and the findings certainly have merit. However, there are certain caveats in the manuscript, mainly in the writing that needs addressing before it can be considered for publication.

We the reviewer for his/her appreciation of the significance and novelty of the work, and for the constructive comments, which have been addressed by further experiments and redrafting as shown below.

Some of my major concerns/comments are listed below:

1. The manuscript fails to provide critical statistical information such as sample size, statistical significance to the differences in measurements acquired etc. It also falls short on indicating the future direction of the study or the impact of these findings on ongoing ATPS researches.

We thank the reviewer for this comment. The evaporation induced non-associative phase separation and the pattern evolution in our model system is robust for sessile droplet with the volume from 0.5 μL to 5 μL , as shown in **Figure 1** (0.5 μL droplet), **Figure 3** (and **Fig. S8**; 0.5 μL droplet loaded with DNA or RNA) and **Figure 5** (5 μL droplet) in the main text, also in **Fig. S9, S10** and **S13** (5 μL droplets loaded with RNA reactants). Besides, as mentioned in the reply to the comment 6 of the reviewer 2, the phase separation phenomenon is also robust for sessile droplets composed of PEG and dextran with a series of different molecular weight (Figure. R1). For the experimental data presented in the main manuscript and supporting information, including radius of phase separation front (R_2 in **Figure 2 (d)**), surface tensions (**Fig. S5**), radius of nucleated droplets (R_5 in **Fig. S7 (a)**) and droplet size distribution in **Fig. S7 (b)**) and fluorescence intensity in **Fig. S12**, they are acquired from the averaging of at least 3 different measurements. All error bars represent S.E.M. (standard error of the mean) from at least three independent experiments. We have added the information into the figure captions accordingly.

For future directions and the impact of our findings, we have added descriptions in the **Discussion** part of the main text:

“Further theoretical models based on the interactions of Navier-Stokes equations and Cahn-Hilliard equations are needed to accurately capture time- and composition-dependant phase separation dynamics inside the evaporating droplet.” and

“Compartmentalization via non-associative phase separation of PEG and dextran has been previously reported as a reactor to enhance ribozyme catalysis, where a reaction rate increase of nearly 70-fold was identified. A recent study demonstrated that RNA self-replication is

enhanced 4-fold through similar compartmentalization in the PEG/dextran system.(Ref 77) Meanwhile, our model system is easy to scale up and highly compatible with wet-dry cycles likely commonplace during early evolution.” and

“It is worthwhile to explore new model systems where multiple mechanisms could work synergistically to enhance prebiotically relevant reactions.”.

2. There appears to be a disjoint in the flow of the manuscript. A considerable length of the paper talks in detail about the kinetic pathways of concentration dependent phase separation of polymers but it is not clear how the first half of the paper plays a role in determining the phase compositions used in all the nucleic acid studies.

We agree with the reviewer's assessment. Accordingly, in the revised manuscript, to illustrate how the phase separation pathway plays a role in nucleic acid localization, and to connect the two parts, we have added a discussion on the enhanced partitioning of nucleic acids into dextran-rich phase due to the increase of the tie line length (TLL), which is determined by phase separation pathways.

To be specific, due to the non-uniform evaporation process, local polymer concentrations near the droplet rim quickly surpass the binodal curve, triggering segregative phase separation inside the sessile droplet. The compositions of the phase-separated structures can be determined from the associated phase diagrams, as shown in **Figure 2 (c)**. DNA and RNA molecules localize into these dextran-rich compartments due to partitioning effects. For biomacromolecules like nucleic acids, the partitioning strength in phase-separated ATPS increases with the increasing tie line length (TLL) in the phase diagram¹⁷. Hence, the pathway of phase separation is important to understand the local enrichment of nucleic acids inside the droplet. Based on the pathway shown in **Figure 2 (c)**, the length of the tie line increases with the evaporation time. Hence, after adding fluorescently labelled DNA, the fluorescence signal is localized into dextran-rich compartments, and the ratio of fluorescence intensity in dextran-rich compartments to that of the surrounding PEG-rich phase increases with the evaporation time, as shown in **Figure 3**, indicating enhanced DNA partitioning according to the increased length of the tie line. A more accurate and detailed formulation in determining the phase compositions would require the coupling of the Navier-Stokes equation and Cahn-Hilliard equation, which is highly nonlinear and could only be solved by full numerical simulations. Developing a comprehensive theory that captures these non-linear effects is certainly exciting. It is, however, beyond the scope of this work. We have added the above qualitative analysis and discussion into the revised manuscript, to point this out and shed some light on future research directions.

In the main text, we have now written

“Hence, the microstructure of phase separation and the hydrodynamic flow are strongly coupled inside the evaporating droplet. With the appearance of rich dynamics, a full understanding on the phase behaviour and pattern evolution during droplet evaporation would require the coupling of the Navier-Stokes equation and Cahn-Hilliard equation. However, the resulting high nonlinearity suggests that it could only be solved by full direct numerical simulations. Nevertheless, kinetic pathways of LLPS obtained by spatiotemporal averaging of polymer concentrations due to water loss have revealed the increase of the tie line length (TLL)

due to evaporation. In the PEG/dextran system, an increase of TLL can lead to the increase of the density difference, the interfacial tension, the viscosity of the dextran-rich phase, as well as partitioning strengths of a third species like nucleic acids and proteins into one of the two phases. Hence, with the understanding of the kinetic pathway and the increasing TLL, our model system can be further harnessed to concentrate biopolymers inside phase-separated compartments, and to serve as the reactor that can enhance a series of biochemical reactions.”

(Page 12 of 28) and

“The increase of K indicates an increase of partitioning strength of DNA and RNA inside the dextran-rich compartments during the evaporation. This is in consistence with the increase of the tie line length (TLL) that determined from the kinetic pathways of phase separation (Figure 2(c)).” **(Page 12 of 28).**

Meanwhile, this localization gives an increase in DNA/RNA concentrations inside the compartments, providing a driving force to accelerate reaction rates, like the enhanced ribozyme cleavage reaction described in the main text. To reveal the mechanisms of enhancing the kinetics of ribozyme cleavage by evaporation induced phase separation, we have also introduced a reaction-diffusion model for the ribozyme cleavage reaction, as described in the main text and supporting information (see **equation (3) and (4)** in the main text, as well as **SI Appendix, S4**, also see the reply to the comment 6 from the Reviewer 1). Our model results show that (see **Figure 5 (e)** in the main text and **Fig. S17** in the SI Appendix), owing to the spatial localization and enrichment of the reactants, the dextran-rich compartments formed through segregative phase separation have significant advantages over water droplets in both accelerating and promoting ribozyme cleavage.

a. Did the authors see any difference in the phenomenon like partitioning, transcription or cleavage in regime I vs regime II?

Based on our analysis about phase separation pathways inside the evaporating droplets (as shown in **Figure 2 (c)**), we conclude that the major differences between phase separation patterns in regime I and regime II are caused by the initial compositions of the sessile droplets. Thermodynamically, the phase-separated compartments in the two regimes share similar kinetics during the evaporation process and thus they can be used as the reactors in a similar manner. We have added the partitioning, transcription and ribozyme cleavage inside the evaporating droplet of regime 1 into supporting information (**Fig. S15**). The results show that these reactions all occur inside the dextran-rich compartments, exhibiting similar phenomena with those observed in the evaporating droplet of regime 2, despite the different patterns of the two regimes.

b. What was the ATPS phase concentrations in each of these studies?

In both DNA transcription and ribozyme cleavage experiments, the sessile droplet has a composition of 11 wt% PEG and 2 wt% dextran. We have added the ATPS concentrations used in these experiments to the **Methods** section.

3. The paper needs more discussion on why, despite having very similar evaporation rates, the phase separation might enhance ribozyme cleavage or DNA transcription compared to

evaporating water droplet? Is the increased relative concentration of substrates in Dextran rich compounds comparable or significantly higher than that in single phase evaporation?

The enhanced reaction rates can be explained by two reasons: 1) The local increase in concentrations of nucleic acids inside the dextran-rich compartments due to the partitioning effects; 2) The sequence-length-dependent partitioning strength of DNA/RNA inside the compartments, favouring accumulation of long ribozyme and release of reaction products with short sequences. As mentioned in the reply to comment 2 above, we derive a simple reaction-diffusion model to illustrate the role of RNA partitioning effects in enhancing ribozyme cleavage reactions inside the evaporating ATPS droplet. The length-dependent partitioning strengths can be modelled by introducing RNA partitioning effects of dextran-rich phase and setting different boundary conditions of the reaction-diffusion kinetic model. We find that 1) the time required to reach the maximum productivity in dextran-rich compartments is about 50 times less than that in water droplets; 2) the final reaction productivity in the dextran-rich domain is about 4 times higher than that in water droplets, qualitatively consistent with our experimental observations. Besides the experimental observation of the drying droplets, our PAGE gel analysis of entire ribozyme cleavage reactions over time also indicates an ATPS induced enhancement, as shown in **Fig. S14**.

We have described this in the main text:

“Figure 5 (e) shows the productivity comparison of the ribozyme cleavage reaction inside the domain of dextran-rich compartment and that of water droplet, where it can be found: 1) the time required to reach the maximum productivity in dextran-rich compartments is about 50 times less than that in water droplets; 2) the final reaction productivity in the dextran-rich domain is about 4 times higher than that in water droplets. Therefore, we conclude from the reaction-diffusion model that due to the partitioning and compartmentalization effects, RNA catalysis reactions can be largely enhanced inside the dextran-rich droplets. Note that the model represents an ideal case where 1) only a stationary phase-separated dextran-rich domain in the evaporating ATPS droplet is considered; 2) the supply of substrate to the dextran-rich phase is always constant; and 3) no loss from degradation or RNase is included, and thus there could be an overestimation of the reaction rate inside the dextran-rich compartments. However, the results from the model are qualitatively consistent with our experimental observations, that the fluorescence signal appears much faster and reaches a higher intensity in evaporating ATPS droplets than in water droplet. Hence, we can conclude that the dextran-rich compartments formed through segregative phase separation have significant advantages over water droplets in both accelerating and promoting ribozyme cleavage, which arise from a spatial localization and enrichment of the reactant RNA inside the dextran-rich compartments triggered by phase separation. ”. (Page 19 of 28).

4. The current discussion section reads more like a conclusion and therefore should be designated as such. Moreover, the paper could benefit from a separate discussion focusing on

different ways the phase separation might impact DNA and RNA activities, compare their findings to literature, discuss the impact of their study on contemporary studies and provide implications of these results on further studies.

We thank the reviewer for this critical comment. We have done a literature review on different ways the phase separation might impact DNA and RNA. We have added a summary with quantitative comparison as follows (also see the reply to the comment 4 from the Reviewer 2).

“Compartmentalization via non-associative phase separation of PEG and dextran has been previously reported as a reactor to enhance ribozyme catalysis, where a reaction rate increase of nearly 70-fold was identified. A recent study demonstrated that RNA self-replication is enhanced 4-fold through similar compartmentalization in the PEG/dextran system.(Ref 77) Meanwhile, our model system is easy to scale up and highly compatible with wet-dry cycles likely commonplace during early evolution.

An alternative pathway for prebiotic compartmentalization could be achieved inside the complex coacervates that formed through associative phase separation of oppositely charged species. RNA partitioning into the coacervates would be mainly achieved by ion-pairing interactions. However, the highly negatively charged nature of RNA indicates that there would always exist complex RNA-polycation interactions that may be unproductive for RNA folding inside the coacervates. Previous studies have identified and tested a variety of complex coacervates formed by polyelectrolytes with different size and sequences. The impact of these coacervates on the prebiotically relevant processes, including RNA partitioning, RNA structural stabilities and enzymatic reactions, varies and largely relies on the physicochemical properties of charged molecules that trigger phase separation. For example, partitioning coefficient of poly U15 inside the poly U-spermine coacervate is about 60-fold less than that of poly A15 RNA. Compared with the reaction in the buffer with no coacervate formed, a rate reduction from 13-fold to 60-fold of hammerhead ribozyme inside polylysine/carboxymethyl-dextran coacervates was found. In contrast, a 5- to 10-time higher product of hammerhead ribozyme cleavage was identified inside the polydiallyldimethylammonium chloride (PDAC) /poly A11 RNA (rA11) coacervates. Hence, more work on the generalization of impact of complex coacervates on RNA folding and reaction rates is needed to fully understand how coacervates formed through associative phase separation could serve as primitive compartments. There may also exist a synergistical mechanism, where both associative and non-associative phase separation could play roles in prebiotic compartmentalization. For example, by the addition of PEG, the RNA oligomer partitioning into the polyU-spermine coacervates was increased by more than two orders of magnitude. Likewise, a more than 5-fold enhancement of transcription rate was achieved inside PEG-containing coacervates of cell lysate protein. It is worthwhile to explore new model systems where multiple mechanisms could work synergistically to enhance prebiotically relevant reactions.”

Some additional minor comments are as follows:

1. It would be easy for the readers to see the fitness of theoretical models if figures S2 and S4 also plotted theoretical and (some) measured values respectively.

We have added the fitness of theoretical results into **Figure S2**. We plot **Figure S4** to illustrate the highly non-uniform evaporation flux near the rim of the sessile droplet. Note that the last data point in Figure S4 denote the total evaporation rate of the sessile droplet, about 0.90 $\mu\text{g/s}$. This value is quite close to the experimentally measured evaporation rate, which is 1.06 (± 0.03) $\mu\text{g/s}$. We have added this comparison in the caption of **Figure S4**.

2. The authors should be attentive not to miss defining all the variables use in the equations and their derivations. (E.g. some variables like τ , RH were missed)

We thank the reviewer for pointing this out. In the revised manuscript, we have given the definitions of these parameters. Specifically, τ is the integral variable and RH is the relative humidity of the experimental condition.

3. What do the pink squares in fig 2 (a) represent?

The circle composed of pink squares (dotted line) represents the location of phase separation front (PSF), which is the boundary of phase-separated region and single-phase region inside the evaporating sessile droplet. We have modified the dotted line to solid line and revised the content in the figure caption and the corresponding main texts.

4. In supplementary text S2, figure S4 is mis-indicated as figure S5.

We think the reviewer for carefully reading our manuscript and catching this typo, which has now been fixed.

5. At that time point was the droplet size and count measured in fig S6 (b)? Weren't the drops coalescing and thus increasing in size with time?

Figure R2. (a) A snapshot of an evaporating droplet in regime 1, where phase-separated PEG-rich droplets are dispersed inside the dextran-rich region; (b) A snapshot of an evaporating droplet in regime 2, where phase-separated dextran-rich droplets are moving towards the droplet centre.

We measured droplet size at a well-defined time point close to the late stage of the evaporation process. Specifically, for the droplet in regime 1, we measure the size distribution of phase-separated PEG-rich droplets inside the dextran-rich region, just before they finally coalesce, as shown in Figure R2 (a). For the droplet in regime 2, we measure the size distribution of phase-separated dextran-rich droplets just before the phase separation front (PSF) moves to the center of the sessile droplet, as shown in Figure R2 (b). We have added the information into the caption of Figure S6 (now **Fig. S7** in the revised SI Appendix).

As pointed out by the reviewer, these organelle-sized droplets will gradually coalesce into bigger ones, as shown in **Fig. S7 (a)**, where we measure the increasing size of phase-separated droplets due to coalescence. As these droplets are highly dynamic, to better understand and harness their dynamic behaviours, it is crucial to understand the kinetics of these phase-separated structures (Angew. Chem. Int. Ed. 2019, **58**, 14489–14494)¹⁸. Our results show that the size distributions of these small droplets, both PEG-rich droplets and dextran-rich droplets, follows an exponential decay distribution, indicating a similar kinetic shared by droplets in

both regimes. Although we cannot provide a more detailed and conclusive exploration, we have decided to report these experimental observations to encourage further research on the coarsening of phase-separated droplets as well as the relevant composition-dependent dynamics inside the evaporating droplet. We believe our work lays a solid foundation for related future research.

Reference

1. Fulton, A.B. How crowded is the cytoplasm? *Cell* **30**, 345-347 (1982).
2. van den Berg, J., Boersma, A.J. & Poolman, B. Microorganisms maintain crowding homeostasis. *Nature Reviews Microbiology* **15**, 309-318 (2017).
3. Gupta, S.K. & Guo, M. Equilibrium and out-of-equilibrium mechanics of living mammalian cytoplasm. *Journal of the Mechanics and Physics of Solids* **107**, 284-293 (2017).
4. Kasza, K.E. & Zallen, J.A. Dynamics and regulation of contractile actin–myosin networks in morphogenesis. *Current Opinion in Cell Biology* **23**, 30-38 (2011).
5. Lawrence, J.B. & Singer, R.H. Intracellular localization of messenger RNAs for cytoskeletal proteins. *Cell* **45**, 407-415 (1986).
6. Ladouceur, A.-M., *et al.* Clusters of bacterial RNA polymerase are biomolecular condensates that assemble through liquid–liquid phase separation. *Proceedings of the National Academy of Sciences* **117**, 18540 (2020).
7. De Gennes, P.-G. & Gennes, P.-G. *Scaling concepts in polymer physics*, (Cornell university press, 1979).
8. Mace, C.R., *et al.* Aqueous Multiphase Systems of Polymers and Surfactants Provide Self-Assembling Step-Gradients in Density. *Journal of the American Chemical Society* **134**, 9094-9097 (2012).
9. Chao, Y. & Shum, H.C. Emerging aqueous two-phase systems: from fundamentals of interfaces to biomedical applications. *Chemical Society Reviews* **49**, 114-142 (2020).
10. Elcock, A.H. Models of macromolecular crowding effects and the need for quantitative comparisons with experiment. *Current Opinion in Structural Biology* **20**, 196-206 (2010).
11. Dewey, D.C., Strulson, C.A., Cacace, D.N., Bevilacqua, P.C. & Keating, C.D. Bioreactor droplets from liposome-stabilized all-aqueous emulsions. *Nat Commun* **5**, 4670 (2014).
12. Fares, H.M., Marras, A.E., Ting, J.M., Tirrell, M.V. & Keating, C.D. Impact of wet-dry cycling on the phase behavior and compartmentalization properties of complex coacervates. *Nat Commun* **11**, 5423 (2020).
13. Tan, H., *et al.* Evaporation-triggered microdroplet nucleation and the four life phases of an evaporating Ouzo drop. *Proceedings of the National Academy of Sciences of the United States of America* **113**, 8642-8647 (2016).
14. Okuzono, T., Ozawa, K.y. & Doi, M. Simple Model of Skin Formation Caused by Solvent Evaporation in Polymer Solutions. *Physical Review Letters* **97**, 136103 (2006).
15. Kajiya, T., Nishitani, E., Yamaue, T. & Doi, M. Piling-to-buckling transition in the drying process of polymer solution drop on substrate having a large contact angle. *Physical Review E* **73**, 011601 (2006).
16. Balsara, N.P., Lin, C. & Hammouda, B. Early Stages of Nucleation and Growth in a Polymer Blend. *Physical Review Letters* **77**, 3847-3850 (1996).
17. Walter, H. *Partitioning in aqueous two–phase system: theory, methods, uses, and applications to biotechnology*, (Elsevier, 2012).
18. Linsenmeier, M., *et al.* Dynamics of Synthetic Membraneless Organelles in Microfluidic Droplets. *Angew Chem Int Ed Engl* **58**, 14489-14494 (2019).

REVIEWERS' COMMENTS

Reviewer #1 (Remarks to the Author):

The authors have responded well to the reviewer comments and the manuscript has been improved considerably. It is recommended to accept the manuscript for publication in Nature Communications.

Reviewer #2 (Remarks to the Author):

The revised manuscript from Guo et al. generally addresses the key concerns of this reviewer. The manuscript now includes text that addresses the role of interfaces in the system, and the manuscript also includes a diffusion-reaction model that captures the enhanced kinetics of the non-associative phase separated micro droplets. With these changes, the work more convincingly conveys the central arguments.

Some minor comments are included below where the authors may have misconstrued the original comments.

Comments:

- The proposed model that the authors employed is simple and sensible. Despite the idealized nature of the model, it appear to capture the enhanced kinetics of the RNA catalysis reactions. The authors state that the "results from the model are quantitatively consisted" with the experiments, but a time-dependent comparison between the model and the data is not apparent.
- The new discussion section is improved, and the new text effectively compares rates observed in similar systems. That said, the goal of the original comment from Reviewer #2 that "The discussion section is more of a summary" was to remove the summary-like text and change the text to read more like a discussion (not to add a summary). This was generally accomplished in the new discussion.
- The range of molecular weights of the PEG and dextran molecules were included. Additionally, the robustness of the system to changes in molecular weight was described, but the original comment was to include a polydispersity index, not just a range and an average.

Reviewer #3 (Remarks to the Author):

The authors have addressed all my queries and concerns. With the addition of significant details and a comprehensive discussion section, the manuscript is now a cogent read for the experts as well as non-experts in the field. Given the novelty and relevance of the findings in this study, I think the manuscript can now be accepted to be published as is.

REVIEWERS' COMMENTS

Reviewer #1 (Remarks to the Author):

The authors have responded well to the reviewer comments and the manuscript has been improved considerably. It is recommended to accept the manuscript for publication in Nature Communications.

Response: We thank the reviewer for his/her enthusiasm and support for publishing this paper in Nature Communications.

Reviewer #2 (Remarks to the Author):

The revised manuscript from Guo et al. generally addresses the key concerns of this reviewer. The manuscript now includes text that addresses the role of interfaces in the system, and the manuscript also includes a diffusion-reaction model that captures the enhanced kinetics of the non-associative phase separated micro droplets. With these changes, the work more convincingly conveys the central arguments.

Response: We thank the reviewer for his/her helpful comments and approval for publishing this work.

Some minor comments are included below where the authors may have misconstrued the original comments.

Comments:

- The proposed model that the authors employed is simple and sensible. Despite the idealized nature of the model, it appear to capture the enhanced kinetics of the RNA catalysis reactions. The authors state that the “results from the model are quantitatively consisted” with the experiments, but a time-dependent comparison between the model and the data is not apparent.

Response: This is indeed a relevant comment. It was stated in our manuscript that “However, the results from the model are *qualitatively* consistent with our experimental observations that the fluorescence signal appears much faster and reaches a higher intensity in evaporating ATPS droplets than in water droplet.” (Page 19/28). Our statement is based on the fact that it is difficult to quantitatively compare the experimental results with the theoretical results, as there might not be a linear response between fluorescence intensity and chemical concentrations in our experiments. Nevertheless, our model can qualitatively capture key mechanisms of enhanced reaction rate through partitioning and compartmentalization during droplet evaporation process. We thank the reviewer for the helpful comment.

- The new discussion section is improved, and the new text effectively compares rates observed in similar systems. That said, the goal of the original comment from Reviewer #2 that “The discussion section is more of a summary” was to remove the summary-like text and change the text to read more like a discussion (not to add a summary). This was generally accomplished in the new discussion.

Response: The reviewer’s helpful comments have clarified the significance of our work. We have removed more summary-like text such as “We quantitatively characterized the pathways of LLPS that determine the phase separation patterns. The surface tension differences induced by non-uniform

evaporation flux drive the movement of LLPS front and the dynamics of nucleated compartments.”, to make the discussion part more concise.

- The range of molecular weights of the PEG and dextran molecules were included. Additionally, the robustness of the system to changes in molecular weight was described, but the original comment was to include a polydispersity index, not just a range and an average.

Response: The reviewer’s comments have indeed helped clarify on the information about the chemicals used. In this study, PEG and dextran are purchased and used without any further processing procedures. The polydispersity index (PDI) of PEG is 1.22, and dextran has the PDI with a range of 1.2-1.7. We have added this information to the Method part in the main manuscript.

Reviewer #3 (Remarks to the Author):

The authors have addressed all my queries and concerns. With the addition of significant details and a comprehensive discussion section, the manuscript is now a cogent read for the experts as well as non-experts in the field. Given the novelty and relevance of the findings in this study, I think the manuscript can now be accepted to be published as is.

Response: We thank the reviewer for his/her enthusiasm and support for publishing this paper in Nature Communications